# Functional Rényi Differential Privacy for Generative Modeling

**Dihong Jiang[1,2],    Sun Sun[1,3]    and    Yaoliang Yu[1,2]**
School of Computer Science, University of Waterloo[1]
Vector Institute[2]
National Research Council Canada[3]
`{dihong.jiang,sun.sun,yaoliang.yu}@uwaterloo.ca`

## Abstract

Differential privacy (DP) has emerged as a rigorous notion to quantify data privacy. Subsequently, Rényi differential privacy (RDP) has become an alternative to the ordinary DP notion in both theoretical and empirical studies, because of its convenient compositional rules and flexibility. However, most mechanisms with DP (RDP) guarantees are essentially based on randomizing a fixed, finite-dimensional vector output. In this work, following Hall et al. [12] we further extend RDP to functional outputs, where the output space can be infinite-dimensional, and develop all necessary tools, e.g. (subsampled) Gaussian mechanism, composition, and post-processing rules, to facilitate its practical adoption. As an illustration, we apply functional RDP (f-RDP) to functions in the reproducing kernel Hilbert space (RKHS) to develop a differentially private generative model (DPGM), where training can be interpreted as iteratively releasing loss functions (in an RKHS) with DP guarantees. Empirically, the new training paradigm achieves a significant improvement in privacy-utility trade-off compared to existing alternatives, especially when $\epsilon = 0.2$. Our code is available at `https://github.com/dihjiang/DP-kernel`.

## 1   Introduction

Modern machine learning has achieved impressive success thanks to the availability of big data and computing resources. However, there are increasing privacy concerns when training with personal or sensitive data. Differential privacy [DP, 8] has become the de-facto standard technique for releasing statistics of sensitive databases, which is designed to bound the output change of a randomized mechanism $\mathcal{M}$ given an incremental input deviation, such that $\mathcal{M}$ does not depend too much on any individual point in the dataset. Recently, Mironov [23] generalizes DP to Rényi differential privacy (RDP) through $\alpha$-Rényi divergence [26], which shares many properties with the ordinary DP, yet with tighter and easier composition analysis, thereby attracts more attention in practice.

The popular mechanisms (e.g. Gaussian or Laplace) towards DP or RDP essentially randomize a finite-dimensional vector output with (Gaussian or Laplace) noises. However, if we want to privately release a *function*, vector-based DP mechanisms are not readily amenable, because a function over a real-valued domain is characterized by infinitely many points [12], which will lead to $\infty$ bound on the L2/L1-sensitivity if we follow the same sensitivity analysis in Dwork and Roth [9] for the vector output, and will effectively lead to infinitely large Gaussian/Laplacian noise and is impractical. Moreover, it is unclear how to add noise to an infinite-dimensional output. Examples that require a functional DP mechanism include privately releasing the reward function in reinforcement learning [32], the kernel function in kernel density estimation and kernel support vector machine (SVM) [12], and the kernel function in DPGM in this work.

37th Conference on Neural Information Processing Systems (NeurIPS 2023).

Hall et al. [12] made the most fundamental contribution to extending DP from vectors to functions. Essentially, the functional Gaussian mechanism is achieved by adding a sample path of Gaussian process to a function, in contrast to adding Gaussian noise to a vector. Evaluating the released DP function at arbitrarily many points (which will form a vector) will retain the same DP guarantee. It is worth mentioning that there are no composition theorems and subsampled Gaussian mechanisms developed for functional DP in Hall et al. [12], thus restricting its use in deep learning.

Due to the theoretical convenience and practical flexibility of RDP, in this work, we aim to extend RDP to functions, with all necessary tools to facilitate its adoption in deep learning. Furthermore, we demonstrate its value via a particular application in DPGM where the loss function is in an RKHS. Our contributions can be summarized as:

- Theoretically, we develop the functional RDP, which is equipped with many useful tools including (subsampled) Gaussian mechanism, composition, and post-processing theorems. We will show that functional RDP will share many important properties and results with the vector-based variant.

- Empirically, with functional RDP, we propose a novel DPGM training paradigm by privatizing the loss function in an RKHS, rather than truncating the RKHS to a finite-dimensional space and injecting Gaussian noise therein as in existing works. Our method is evaluated and compared across a wide variety of image datasets and DP guarantees, where our method consistently outperforms other baselines by a large margin. Notably, our method indicates better scalability at more stringent DP guarantees (e.g., $\epsilon = 1$ and $0.2$), compared to state-of-the-art (SoTA) baselines.

## 2   Preliminary

Differential privacy quantifies and restricts the output change of a randomized mechanism given an incremental change in the input dataset, such that the privacy of an individual point is protected. The output of a randomized mechanism can be either a vector (Sections 2.1 to 2.3) or a function (Section 2.4). In this section, we recall a few important related works in differential privacy.

### 2.1   Differential privacy for vectors (v-DP)

**Definition 1** (($\epsilon, \delta$)-DP for vectors, [8, 9]). *A randomized mechanism $\mathcal{M} : \mathcal{D} \to \mathcal{R}$ with domain $\mathcal{D}$ and range $\mathcal{R}$ satisfies ($\epsilon, \delta$)-differential privacy if for any two adjacent inputs $D, D' \in \mathcal{D}$ and for any (measurable) subset of outputs $\mathcal{S} \subseteq \mathcal{R}$ it holds that*

$$\Pr[\mathcal{M}(D) \in \mathcal{S}] \leq \exp(\epsilon) \cdot \Pr[\mathcal{M}(D') \in \mathcal{S}] + \delta,$$

*where adjacent inputs (a.k.a. neighbouring datasets) $D, D'$ only differ in one entry. Particularly, when $\delta = 0$, we say that $\mathcal{M}$ is $\epsilon$-DP.*

### 2.2   Rényi differential privacy for vectors (v-RDP)

Mironov [23] first formalizes Rényi differential privacy (RDP) which extends ordinary DP by using $\alpha$-Rényi divergence [26]. RDP is shown to provide easier composition properties than the ordinary DP notion, and it can be easily converted to ($\epsilon, \delta$)-DP.

**Definition 2** (($\alpha, \epsilon$)-RDP for vectors, [23]). *A randomised mechanism $\mathcal{M}$ is ($\alpha, \epsilon$)-RDP if for all adjacent inputs $D, D'$, Rényi's $\alpha$-divergence (of order $\alpha > 1$) between the distributions of $\mathcal{M}(D)$ and $\mathcal{M}(D')$ satisfies:*

$$\mathbb{D}_\alpha(\mathcal{M}(D)\|\mathcal{M}(D')) := \tfrac{1}{\alpha-1} \log \mathbb{E}_{x \sim q} \left( \tfrac{p(x)}{q(x)} \right)^\alpha \leq \epsilon,$$

*where $p$ and $q$ are the density of $\mathcal{M}(D)$ and $\mathcal{M}(D')$, respectively.*

Conveniently, RDP is linearly composable:

**Theorem 1** (Sequential composition of v-RDP, [23]). *Let $f : \mathcal{D} \to \mathcal{R}_1$ be ($\alpha, \epsilon_1$)-RDP, $g : \mathcal{R}_1 \times \mathcal{D} \to \mathcal{R}_2$ be ($\alpha, \epsilon_2$)-RDP, then running $f, g$ sequentially to obtain $h : \mathcal{D} \to \mathcal{R}_1 \times \mathcal{R}_2, h(D) := \big(f(D), g(f(D), D)\big)$ satisfies ($\alpha, \epsilon_1 + \epsilon_2$)-RDP.*

Similar to the parallel composition theorem for ordinary $\epsilon$-DP as in McSherry [22], we complement the parallel composition for v-RDP:

**Theorem 2** (Parallel composition of v-RDP). *If mechanism $\mathcal{M}_i$ satisfies $(\alpha, \epsilon_i)$-RDP for $i = 1, 2, \ldots, m$, and let $D_1, D_2, \ldots, D_m$ be the disjoint partitions by executing a deterministic partitioning function $P$ on $D$. Releasing $\mathcal{M}_1(D_1), \ldots, \mathcal{M}_m(D_m)$ satisfies $(\alpha, \max_{i \in \{1,2,\ldots,m\}} \epsilon_i)$-RDP.*

## 2.3 Gaussian mechanism for v-DP and v-RDP

Among multiple choices, the Gaussian mechanism is more suitable for the $(\epsilon, \delta)$-DP notion (where $\delta > 0$) and provides additional flexibility (e.g., the sum of Gaussians is still a Gaussian). It is achieved by adding calibrated spherical Gaussian noise to a vector output.

**Proposition 1** (Gaussian mechanism for v-DP and v-RDP, [9, 23]). *Given a $d$-dimensional function $f : \mathcal{D} \to \mathbb{R}^d$. The Gaussian mechanism is given by:*
$$\mathcal{M}(D) = f(D) + \sigma \Delta_2 f \cdot \mathcal{N}(0, \mathbb{I}_d),$$
*where $\Delta_2 f = \max_{D \sim D', D, D' \in \mathcal{D}} \|f(D) - f(D')\|_2$. $\mathcal{M}(D)$ is said to be: (1) $(\epsilon, \delta)$-DP if $\sigma \geq \sqrt{2 \ln(1.25/\delta)}/\epsilon$ for $\epsilon \in (0, 1)$, or (2) $(\alpha, \frac{\alpha}{2\sigma^2})$-RDP.*

## 2.4 Differential privacy for functions (f-DP)

Prior DP works mainly focused on the setting when the output of a query is a vector. However, if the output is a *function*, such as the kernel density estimation example in Hall et al. [12] and the DPGM example in our work (Section 4.3), a few challenges render the direct application of previous DP results on a *functional output*[1] intractable: it is unclear (1) how to analyze the sensitivity and (2) how to add Gaussian noise to an infinite-dimensional output. To the best of our knowledge, Hall et al. [12] are the pioneers in addressing those challenges by starting with defining the $(\epsilon, \delta)$-DP for functions:

**Definition 3** ($(\epsilon, \delta)$-DP for functions, [12]). *Consider a class of functions indexed by database $D$ over $T = \mathbb{R}^d$, i.e. $\{f_D : D \in \mathcal{D}\} \subseteq \mathbb{R}^T$. Define cylinder sets[2] $C_{S,B} = \{f \in \mathbb{R}^T : (f(\mathbf{x}_1), \ldots, f(\mathbf{x}_n)) \in B\}$, for all finite subsets $S = (\mathbf{x}_1, \ldots, \mathbf{x}_n)$ of $T$ and Borel sets[3] $B \in \mathbb{R}^n$. Then, define $\mathcal{C}_s = \{C_{S,B} : B \in \mathcal{B}(\mathbb{R}^n)\}$ and $\mathcal{F}_0 = \bigcup_{S:|S|<\infty} \mathcal{C}_s$. We say the mechanism $\widetilde{f_D}$ satisfies $(\epsilon, \delta)$-DP over the field of cylinder sets, if for all $D, D' \in \mathcal{D}$:*
$$\Pr[\widetilde{f_D} \in A] \leq \exp(\epsilon) \times \Pr[\widetilde{f_{D'}} \in A] + \delta, \quad \forall A \in \mathcal{F}_0. \tag{1}$$

Hall et al. [12] point out that whenever Eq. (1) holds, for any finite set of points $\mathbf{x}_1, \ldots, \mathbf{x}_n$ in $T$ chosen a-priori, or after the construction of the function, or even adaptively chosen based on the outputs given, releasing the vector $[\widetilde{f_D}(\mathbf{x}_1), \ldots, \widetilde{f_D}(\mathbf{x}_n)]$ satisfies $(\epsilon, \delta)$-DP.

The Gaussian mechanism for f-DP [4] is reached by injecting a sample path of a calibrated Gaussian process[5] into the function:

**Proposition 2** (Gaussian mechanism for f-DP, [12]). *Let $G$ be a sample path of a Gaussian process having mean zero and covariance function $k$. Let $M$ denote the Gram matrix*
$$M(\mathbf{x}_1, \ldots, \mathbf{x}_n) = \begin{pmatrix} k(\mathbf{x}_1, \mathbf{x}_1) & \ldots & k(\mathbf{x}_1, \mathbf{x}_n) \\ \vdots & \ddots & \vdots \\ k(\mathbf{x}_n, \mathbf{x}_1) & \ldots & k(\mathbf{x}_n, \mathbf{x}_n) \end{pmatrix}. \tag{2}$$

*Let $\{f_D : D \in \mathcal{D}\}$ be a family of functions indexed by database $D$. Releasing $\widetilde{f_D} = f_D + \sqrt{2 \ln(1.25/\delta)} \Delta/\epsilon \cdot G$ satisfies $(\epsilon, \delta)$-DP whenever*
$$\sup_{D \sim D'} \sup_{n < \infty} \sup_{(\mathbf{x}_1, \ldots, \mathbf{x}_n) \in T^n} \left\| M^{-\frac{1}{2}}(\mathbf{x}_1, \ldots, \mathbf{x}_n) \begin{pmatrix} f_D(\mathbf{x}_1) - f_{D'}(\mathbf{x}_1) \\ \vdots \\ f_D(\mathbf{x}_n) - f_{D'}(\mathbf{x}_n) \end{pmatrix} \right\|_2 \leq \Delta. \tag{3}$$

---

[1] It means the output is a function.

[2] Intuitively, the cylinder set defined here is a set of functions where any finite evaluation of the function is in a "ball".

[3] A Borel set is formed via countable union, intersection and complement of open sets in a topological space.

[4] Not to confuse with the function DP in Dong et al. [7], where the type-I and type-II errors are controlled by a pre-specified function.

[5] A Gaussian process is a generalization of the multivariate Gaussian distribution to the infinite-dimensional case. Any arbitrarily collected finite random variables from a Gaussian process follow a multivariate Gaussian distribution.

Particularly, Hall et al. [12] studied how to achieve $(\epsilon, \delta)$-DP for functions in an RKHS [6] $\mathcal{H}$:

**Corollary 1** (Corollary 9 in [12]). *For $\{f_D : D \in \mathcal{D}\} \subseteq \mathcal{H}$, releasing $\widetilde{f_D} = f_D + \sqrt{2 \ln(1.25/\delta)} \Delta/\epsilon \cdot G$ is $(\epsilon, \delta)$-DP (with respect to the cylinder $\sigma$-field) whenever $\sup_{D,D'} \|f_D - f_{D'}\|_{\mathcal{H}} \leq \Delta$ and when $G$ is a sample path of a Gaussian process with mean zero and covariance function $k$ that is given by the reproducing kernel of $\mathcal{H}$.*

# 3 Rényi differential privacy for functions (f-RDP)

In this section, we aim to extend the definition of RDP to functional outputs, along with its associated calculus rules to facilitate practical adoption. In particular, we will show that the main results for v-RDP all extend to f-RDP.

## 3.1 Definition

Consider a class of functions over $T = \mathbb{R}^d$, i.e. $\{f_D : D \in \mathcal{D}\} \subseteq \mathbb{R}^T$. We can define a functional generalization of the $\alpha$-Rényi divergence:

**Definition 4** $((\alpha, \epsilon)$-RDP for functions). *Denote the evaluation of function $\widetilde{f_D}$ on any finite subsets $S = (\mathbf{x}_1, \ldots, \mathbf{x}_n)$ of $T$ by $\{\widetilde{f_D}(\mathbf{x}_1), \ldots, \widetilde{f_D}(\mathbf{x}_n)\} := \widetilde{f_D}(S)$. We say $\widetilde{f_D}$ is $(\alpha, \epsilon)$-RDP, if for all adjacent inputs $D, D' \in \mathcal{D}$, Rényi's $\alpha$-divergence (of order $\alpha > 1$) between the distributions of $\widetilde{f_D}(S)$ and $\widetilde{f_{D'}}(S)$ satisfies:*

$$\mathbb{D}_\alpha\left(\widetilde{f_D}(S)\|\widetilde{f_{D'}}(S)\right) := \frac{1}{\alpha-1} \log \mathbb{E}_{x \sim q}\left(\frac{p(x)}{q(x)}\right)^\alpha \leq \epsilon, \tag{4}$$

*where $p, q$ are the density of $\widetilde{f_D}(S)$ and $\widetilde{f_{D'}}(S)$, respectively.*

**Remark 1.** *Definition 4 essentially claims that the distribution of any finite number of evaluations of function $\widetilde{f_D}$ and $\widetilde{f_{D'}}$ satisfies Definition 2 (v-RDP).*

We have a weaker definition of f-RDP (see Lemma 1 for the *weaker* definition) also based on cylinder sets as in Definition 3. We also refer interested readers to Appendix F for an alternative unified definition of RDP for both vectors and functions.

## 3.2 Post-processing theorem

As any data-independent post-processing preserves the DP guarantee for v-RDP (Definition 2), it also preserves DP guarantee for f-RDP with Remark 1. Specifically,

**Theorem 3** (Post-processing theorem of f-RDP). *If a function $f_D$ is $(\alpha, \epsilon)$-RDP, so is $g \circ f_D$, where $g$ is a post-processing mechanism that only depends on a finite number of outputs of $f_D$.*

## 3.3 Conversion to $(\epsilon, \delta)$-DP

The following conversion is a direct extension of its vector version in Mironov [23]:

**Proposition 3** (f-RDP conversion to f-DP). *A function $\widetilde{f_D}$ that is $(\alpha, \epsilon)$-RDP is $(\epsilon + \frac{\log 1/\delta}{\alpha-1}, \delta)$-DP.*

## 3.4 Composition theorems

Similar to Theorem 2, we first derive the parallel composition theorem of RDP for functions:

**Theorem 4** (Parallel composition of f-RDP). *Given a deterministic partitioning function $P$, let $D_1, D_2, \ldots, D_m$ be the disjoint partitions by executing $P$ on $D$. If function $f_{D_i}$ satisfies $(\alpha, \epsilon_i)$-RDP for $i = 1, 2, \ldots, m$, releasing $(f_{D_1}, \ldots, f_{D_m}) := f_D$ satisfies $(\alpha, \max_{i \in \{1,2,\ldots,m\}} \epsilon_i)$-RDP.*

Now we move on to the sequential composition theorem of RDP for functions (extension of Theorem 1 to functional mechanisms), which is important and required for composing the total privacy cost when we sample different sample paths from the Gaussian process over training iterations.

---

[6] A Hilbert space of functions with continuous pointwise evaluation $f \mapsto f(x)$. We give more explanations in Section 4.1.

First, we note that Theorem 1 implicitly defines $g$ as a functional mechanism, i.e. $g : \mathcal{D} \to \mathcal{R}_2^{\mathcal{R}_1}$. It means for any $r_1 \in \mathcal{R}_1$, we have $g_{r_1} := g(r_1, \cdot) : \mathcal{D} \to \mathcal{R}_2$ is $(\alpha, \epsilon_2)$-RDP.

**Theorem 5** (Sequential composition of f-RDP). *Let $\{f_D : D \in \mathcal{D}\}$ and $\{g_D : D \in \mathcal{D}\}$ be two families of functions indexed by dataset $D$, where $f_D \in \mathcal{R}_1^T$ is $(\alpha, \epsilon_1)$-RDP and $g_D : \mathcal{R}_1^T \to \mathcal{R}_2^S$ is $(\alpha, \epsilon_2)$-RDP. Releasing the sequentially composed functional mechanism $h_D = (f_D, g_D \circ f_D) \in \mathcal{R}_1^T \times \mathcal{R}_2^S = (\mathcal{R}_1 \times \mathcal{R}_2)^{T \times S}$ satisfies $(\alpha, \epsilon_1 + \epsilon_2)$-RDP.*

### 3.5 Gaussian mechanism

We also need to develop a Gaussian mechanism to retain f-RDP. To make it more convenient, we first rewrite Proposition 1 in a similar form to Proposition 3 in Hall et al. [12] as:

**Definition 5** (Gaussian mechanism for v-RDP, with non-isotropic Gaussian). *Let $M \in \mathbb{R}^{d \times d}$ be a positive definite symmetric matrix, the family of vectors $\{\mathbf{v}_D : D \in \mathcal{D}\} \in \mathbb{R}^d$ satisfies $\sup_{D \sim D'} \|M^{-\frac{1}{2}}(\mathbf{v}_D - \mathbf{v}_{D'})\|_2 \leq \Delta$ for all adjacent datasets $D, D' \in \mathcal{D}$. The Gaussian mechanism is given by:*

$$\widetilde{\mathbf{v}_D} = \mathbf{v}_D + \sigma \Delta \cdot \mathcal{N}(0, M). \tag{5}$$

$\widetilde{\mathbf{v}_D}$ in Eq. (5) satisfies $(\alpha, \frac{\alpha}{2\sigma^2})$-RDP. Now we define the Gaussian mechanism for f-RDP:

**Proposition 4** (Gaussian mechanism for f-RDP). *Let $G$ be a sample path of a Gaussian process having mean zero and covariance function $k$. Let $M$ denote the Gram matrix (as defined in Eq. (2)). Let $\{f_D : D \in \mathcal{D}\}$ be a family of functions indexed by database $D$. Releasing $\widetilde{f_D} = f_D + \sigma \Delta \cdot G$ satisfies $(\alpha, \frac{\alpha}{2\sigma^2})$-RDP whenever Eq. (3) holds.*

Particularly, when the function is in an RKHS $\mathcal{H}$, we have the following corollary:

**Corollary 2.** *For $\{f_D : D \in \mathcal{D}\} \subseteq \mathcal{H}$, releasing $\widetilde{f_D} = f_D + \sigma \Delta \cdot G$ is $(\alpha, \frac{\alpha}{2\sigma^2})$-RDP (with respect to the cylinder $\sigma$-field) whenever $\sup_{D, D'} \|f_D - f_{D'}\|_{\mathcal{H}} \leq \Delta$ and when $G$ is a sample path of a Gaussian process with mean zero and covariance function $k$ given by the reproducing kernel of $\mathcal{H}$.*

### 3.6 Subsampled Gaussian mechanism (SGM)

Subsampling is a crucial component in existing DP deep learning algorithms (e.g., DP-SGD of [1]), which incurs a small privacy cost for every sampled batch by querying $D$, thereby requiring to compose the total privacy loss (measured by $\epsilon$) over training iterations. The same thing could happen for functional mechanisms when we subsample a batch $S$ from the whole dataset $D$ to index the function $f_S$.

The current dominant (python-based) DP implementation packages, e.g., Tensorflow privacy and Opacus, apply v-RDP and compute the total $\epsilon$ in three main steps: (1) compute the v-RDP guarantee for an SGM, based on a numerical procedure in Mironov et al. [24]; (2) sequentially compose v-RDP over training iterations; (3) convert v-RDP to v-DP. In previous sections we have already shown that f-RDP shares the same results with v-RDP on steps (2) and (3), now we will show that an SGM for f-RDP can be reduced to v-RDP in Mironov et al. [24], thus, the numerical procedure in Mironov et al. [24] (and thereby those DP packages) is amenable to f-RDP in terms of the $\epsilon$ accumulation.

We apply the same subsampling strategy as in Mironov et al. [24]. Let $S$ be a subsampled set from $D$, where each element of $S$ is independently drawn from $D$ with probability $q$. SGM for f-RDP is given by $\widetilde{f_S} = f_S + \sigma \Delta \cdot G$, where $\Delta$ and $G$ are defined in Proposition 4. The reduction is immediate: for any finite set of points $V = \{\mathbf{x}_1, \ldots, \mathbf{x}_d\}$ (where $d < \infty$), $f_S(V)$ will form a $d$-dimensional vector. Let $g_S(V) = M^{-1/2} f_S(V)$, we have $\widetilde{g_S}(V) = g_S(V) + \sigma \Delta \cdot \mathcal{N}(0, \mathbb{I}_d)$, where $g_S(V)$ is a $d$-dimensional vector and $\widetilde{g_S}(V)$ is the same SGM as in Mironov et al. [24]. Therefore, f-RDP shares the same guarantee of an SGM with v-RDP. Another intuition is that Mironov et al. [24] reduce computing the $\alpha$-Rényi divergence of $d$-dimensional Gaussians to 1-dimensional Gaussians, which guides all of their subsequent derivations. When $d \to \infty$, we can reach the same reduction from infinite-dimensional Gaussian (Gaussian process) to 1-dimensional Gaussian.

# 4 An application in DPGM

To demonstrate the practical value of f-RDP, here we consider a particular example of training a differentially private generative model (DPGM) through Maximum Mean Discrepancy (MMD).

## 4.1 Background

Assuming a feature map $\phi : \mathcal{X} \to \mathcal{H}$, where $\mathcal{H}$ is an RKHS of some kernel, MMD [11] is a non-parametric distance measure that compares two distributions $p, q$ by:

$$\mathcal{L}_{\mathrm{MMD}^2}(p, q) = \|\mathbb{E}_{\mathbf{x} \sim p}[\phi(\mathbf{x})] - \mathbb{E}_{\mathbf{w} \sim q}[\phi(\mathbf{w})]\|_{\mathcal{H}}^2,$$

where $\mathbb{E}_{\mathbf{x} \sim p}[\phi(\mathbf{x})] \in \mathcal{H}$ is also known as the (kernel) mean embedding (KME) of $p$.

Given a kernel $k : \mathcal{X} \times \mathcal{X} \to \mathbb{R}$, such that[7] $k(\mathbf{x}, \mathbf{w}) = \langle \phi(\mathbf{x}), \phi(\mathbf{w}) \rangle_{\mathcal{H}}$, we can play the kernel trick to compute the (squared) MMD in an alternative way:

$$\mathcal{L}_{\mathrm{MMD}^2}(p, q) = \mathbb{E}_{\mathbf{x}, \mathbf{x}' \sim p} k(\mathbf{x}, \mathbf{x}') - 2\mathbb{E}_{\mathbf{x} \sim p, \mathbf{w} \sim q} k(\mathbf{x}, \mathbf{w}) + \mathbb{E}_{\mathbf{w}, \mathbf{w}' \sim q} k(\mathbf{w}, \mathbf{w}'),$$

which also implicitly lifts the KME into an infinite-dimensional space.

Sriperumbudur et al. [27] suggest that if $k$ is a characteristic kernel (e.g., Gaussian kernel), then MMD $= 0$ iff $p = q$, which makes MMD a practical tool in many applications, such as two-sample test [11] and generative modeling [e.g., 16, 17].

## 4.2 Related works

Balog et al. [3] first proposed a DP database release mechanism via KME, by truncating the infinite-dimensional RKHS to a finite-dimensional feature space through random Fourier features and adding Gaussian noise to the mean of truncated feature embeddings of all data. This idea is further extended to generative modeling by DP-MERF [13]. Specifically, given the samples drawn from two distributions: $D = \{\mathbf{x}_i\}_{i=1}^N \sim p$ (true) and $W = \{\mathbf{w}_j\}_{j=1}^M \sim q$ (generated), empirical MMD with a kernel function $k$ can be estimated by:

$$\widehat{\mathcal{L}_{\mathrm{MMD}^2}}(p, q) = \frac{1}{N^2} \sum_{i=1}^N \sum_{p=1}^N k(\mathbf{x}_i, \mathbf{x}_p) - \frac{2}{NM} \sum_{i=1}^N \sum_{j=1}^M k(\mathbf{x}_i, \mathbf{w}_j) + \frac{1}{M^2} \sum_{j=1}^M \sum_{q=1}^M k(\mathbf{w}_j, \mathbf{w}_q). \quad (6)$$

DP-MERF approximates the kernel by: $k(\mathbf{x}, \mathbf{w}) = \widehat{\phi}(\mathbf{x})^\top \widehat{\phi}(\mathbf{w})$, where $\widehat{\phi}(\mathbf{x}) \in \mathbb{R}^d$ is the random Fourier feature [25] and $d$ is the feature dimension. Now the loss becomes:

$$\widehat{\mathcal{L}_{\mathrm{MMD}_{rf}^2}}(p, q) = \left\| \frac{1}{N} \sum_{i=1}^N \widehat{\phi}(\mathbf{x}_i) - \frac{1}{M} \sum_{j=1}^M \widehat{\phi}(\mathbf{w}_j) \right\|_2^2 = \left\| \widehat{\mu_p} - \frac{1}{M} \sum_{j=1}^M \widehat{\phi}(\mathbf{w}_j) \right\|_2^2,$$

where $\widehat{\mu_p} = \frac{1}{N} \sum_{i=1}^N \widehat{\phi}(\mathbf{x}_i)$ is the empirical KME of $p$. Calibrated Gaussian noise is added to $\widehat{\mu_p}$ to obtain $\widetilde{\mu_p}$ with DP guarantees, which can be viewed as privatized statistics of the whole real dataset. Thereafter, the training objective is to match the KME of generated data with $\widetilde{\mu_p}$, without querying the real data any longer. A line of recent follow-up works, e.g., PEARL [19] and DP-HP [31], boil down to improving the finite-dimensional truncation and show some further improvement in utility.

Compared to other DPGM approaches via DP-SGD [1], DP-MERF is appealing in two aspects: (1) training efficiency, since noise is added to the KME of the whole database once and for all, whereas DP-SGD has to clip and perturb the gradient in each training iteration, which leads to significant training time overhead; (2) more scalable to smaller $\epsilon$, e.g., $\epsilon = 1$ or below.

However, we observe that the generation by DP-MERF (and related followup methods) resembles "mean-like" images (e.g. see Figures 1 to 3), which can be explained by their training objective, because matching the KME of all data is likely to lead to *mode collapse*. Moreover, truncating the RKHS into a finite-dimensional space makes it easy to add Gaussian noise, but at the cost of potentially losing the fine ability to distinguish the data distribution from generation (any finite-dimensional RKHS is not characteristic). Therefore, we turn to study the possibility of adding noise in the infinite-dimensional RKHS directly.

---

[7]By Moore–Aronszajn theorem, any symmetric, positive definite kernel $k$ on $\mathcal{X}$ defines a unique Hilbert space of functions on $\mathcal{X}$ where $k$ is a reproducing kernel. We can find a feature map $\phi : \mathcal{X} \to \mathcal{H}$, such that $k(\mathbf{x}, \mathbf{w}) = \langle \phi(\mathbf{x}), \phi(\mathbf{w}) \rangle_{\mathcal{H}}$ and $\langle \cdot, \cdot \rangle_{\mathcal{H}}$ is the inner product on $\mathcal{H}$.

## 4.3 Methodology

We use Gaussian kernel as our kernel function $k$, i.e., $k(\mathbf{x}, \mathbf{w}) = \exp\{-\frac{\|\mathbf{x}-\mathbf{w}\|_2^2}{2h^2}\}$, since it is widely used in related works [13, 19, 31]. Define

$$f_D = \frac{1}{N} \sum_{i=1}^N \phi(\mathbf{x}_i) = \frac{1}{N} \sum_{i=1}^N k(\mathbf{x}_i, \cdot),$$

so we can rewrite Eq. (6) by plugging in $f_D$:

$$\widehat{\mathcal{L}_{\text{MMD}^2}} = \frac{1}{N} \sum_{p=1}^N f_D(\mathbf{x}_p) - \frac{2}{M} \sum_{j=1}^M f_D(\mathbf{w}_j) + \frac{1}{M^2} \sum_{j=1}^M \sum_{q=1}^M k(\mathbf{w}_j, \mathbf{w}_q).$$

Privatizing the terms relating to real data $\mathbf{x}_i \in D$ ($i = 1, 2, \ldots, N$) amounts to privatizing the function $f_D \in \mathcal{H}$ by Corollary 2, which leads to our private training objective:

$$\widetilde{\mathcal{L}_{\text{MMD}^2}} = \frac{1}{N} \sum_{p=1}^N \widetilde{f_D}(\mathbf{x}_p) - \frac{2}{M} \sum_{j=1}^M \widetilde{f_D}(\mathbf{w}_j) + \frac{1}{M^2} \sum_{j=1}^M \sum_{q=1}^M k(\mathbf{w}_j, \mathbf{w}_q), \tag{7}$$

where $\mathbf{w} = g_\theta(\mathbf{z})$, i.e., $\mathbf{w}$ is generated from standard Gaussian noise $\mathbf{z}$ by a generative neural network $g_\theta$. Given the kernel is Gaussian, we can easily bound the sensitivity of $f_D$ in the RKHS norm. Without loss of generality, assume $D, D'$ only differ in the last element, i.e., $\mathbf{x}_N \neq \mathbf{x}'_N$. Then,

$$(f_D - f_{D'}) = \frac{1}{N} \sum_{i=1}^N \phi(\mathbf{x}_i) - \frac{1}{N} \sum_{i=1}^N \phi(\mathbf{x}'_i) = \frac{1}{N}\big(\phi(\mathbf{x}_N) - \phi(\mathbf{x}'_N)\big).$$

Now we play the kernel trick $k(\mathbf{x}, \mathbf{y}) = \langle \phi(\mathbf{x}), \phi(\mathbf{y}) \rangle_{\mathcal{H}}$ again:

$$\|f_D - f_{D'}\|_{\mathcal{H}}^2 = \frac{1}{N^2}\big(k(\mathbf{x}_N, \mathbf{x}_N) - 2k(\mathbf{x}_N, \mathbf{x}'_N) + k(\mathbf{x}'_N, \mathbf{x}'_N)\big) \leq \frac{2}{N^2} := \Delta^2.$$

We follow *the batch method* in Hall et al. [12] to release function $\widetilde{f_D}$ in practice, as it naturally fits the batch training manner, which amounts to sampling a path from the Gaussian process specified by any finite collection (batch) of points. Assuming the batch size is $m$, we concatenate $(\mathbf{x}, \mathbf{w}) = \mathbf{s}$ (of size $2m$, $\mathbf{x} = \mathbf{s}_{[1:m]}, \mathbf{w} = \mathbf{s}_{[m+1:2m]}$), to save an additional privacy cost incurred by an additional sample path in each training iteration. Now we release $\widetilde{f_D}(\mathbf{x}_i)$ and $\widetilde{f_D}(\mathbf{w}_j)$ via $\widetilde{f_D}(\mathbf{s})$ as needed in Eq.(7) by:

$$\begin{pmatrix} \widetilde{f_D}(\mathbf{s}_1) \\ \vdots \\ \widetilde{f_D}(\mathbf{s}_{2m}) \end{pmatrix} \sim \mathcal{N}\left( \begin{pmatrix} f_D(\mathbf{s}_1) \\ \vdots \\ f_D(\mathbf{s}_{2m}) \end{pmatrix}, \sigma\Delta \begin{pmatrix} k(\mathbf{s}_1, \mathbf{s}_1) & \ldots & k(\mathbf{s}_1, \mathbf{s}_{2m}) \\ \vdots & \ddots & \vdots \\ k(\mathbf{s}_{2m}, \mathbf{s}_1) & \ldots & k(\mathbf{s}_{2m}, \mathbf{s}_{2m}) \end{pmatrix} \right). \tag{8}$$

In summary, each training iteration proceeds with (1) subsample a batch $S$ from $D$ to index the function $f_S$; (2) perturb $f_S$ by Corollary 2; (3) compute the loss function Eq. (7) by Eq. (8).

**Extension to the conditional setting:** We follow the approach in DP-MERF to encode labels in the MMD loss. Consider a new kernel $k^*$ as a product of two existing kernels: $k^*\big((\mathbf{x}, \mathbf{y}), (\tilde{\mathbf{x}}, \tilde{\mathbf{y}})\big) = k_{\mathbf{x}}(\mathbf{x}, \tilde{\mathbf{x}}) k_{\mathbf{y}}(\mathbf{y}, \tilde{\mathbf{y}})$, where we set $k_{\mathbf{x}}$ the same as the unconditional setting (i.e., Gaussian kernel) and $k_{\mathbf{y}}$ to be polynomial kernel of order-1, i.e., $k_{\mathbf{y}}(\mathbf{y}, \tilde{\mathbf{y}}) = \mathbf{y}^\top \tilde{\mathbf{y}}$. Now the function that we want to privately release becomes $f_D^* = \frac{1}{N} \sum_{i=1}^N k^*((\mathbf{x}_i, \mathbf{y}_i), (\cdot, \cdot))$. The sensitivity of $f_D^*$ is the same as $f_D$. Thus, releasing $f_D^*$ by *the batch method* is achieved by replacing $f_D, k$ with $f_D^*, k^*$ in Eq. (8).

## 4.4 Experiments

Since our method is based on making the RKHS features of a (Gaussian) kernel differentially private without truncating the kernel, we term our method **DP-kernel**. In this section, we will evaluate DP-kernel both qualitatively and quantitatively on three image benchmarks, including both grayscale and colorful images. All implementation details are given in Appendix C.

**Datasets:** We consider widely used image benchmarks in related works, i.e. MNIST [15], Fashion MNIST [34], and CelebA [20]. For MNIST and Fashion MNIST, we generate images conditioned on 10 respective labels. For CelebA, we condition on gender. Descriptions and pre-processing of the datasets are given in Appendix B.

**Evaluation metrics:** For quantitative comparison with baselines, we compute the following two metrics via 60k generated images under the same DP guarantees as baselines:

- Generation fidelity. We mainly use Fréchet Inception Distance (FID) [14], as it is widely compared in related works. However, recent studies suggest that FID is not always consistent with human evaluations (e.g. [28]), where the authors found that Inception-V3 (the feature extractor used to compute FID, which was proposed in 2015) is not competent enough, and switching to a more recent and powerful feature extractor (e.g. DINOv2 ViT-L/14) seems to overcome this issue. Therefore, we follow Stein et al. [28] and use their code to compute an additional Fréchet Distance, which is denoted by $FD_{DINOv2}$, for our presentation only.

- Generation utility. We train a convolutional neural network (CNN) as the classifier on generated images, then test the classifier on real images, where the performance is measured by the classification accuracy. We take 5 runs and report the average and standard deviation.

**Baselines:** We compare our method with the following related works: (1) kernel-based methods: DP-MERF [13], DP-HP [31], PEARL [19]; (2) others: DP-CGAN [29], GS-WGAN [5], DP-Sinkhorn [4], G-PATE [21], DataLens [33], DPDM [6]. All baselines are developed from v-RDP or v-DP.

**Privacy regimes:** Note that according to Definition 1, the DP guarantee is weak when $\epsilon \geq 10$, because $\exp(10) \approx 2.2 \times 10^4$, which is meaningless, because the two probabilities are presumed to be comparable for practical deployment (e.g. $\epsilon \leq 1$). However, a line of recent SoTA DPGMs only generate acceptable images at $\epsilon = 10$ [4, 5, 29]. Instead, we consider three values of $\epsilon$, i.e., $10, 1, 0.2$, indicating three levels of DP guarantees, where we put comparison under $(10, 10^{-5})$-DP in Appendix D for completeness.

#### 4.4.1 Conditional vs. parallel

The downstream classification task requires access to labels, so we need to consider conditional generative modeling, as introduced at the end of Section 4.3.

Alternatively, we can partition the dataset by labels, train $K$ ($K$ is the number of total classes) unconditional sub-models in parallel with DP guarantees, and release the union in the end, which could retain the same level of DP guarantee as the conditional one by parallel composition theorem for f-RDP (Theorem 4). Specifically, we partition the dataset $D$ into $K$ subsets by labels, and set the target $\epsilon$ (for $f_D$) the same for all $f_{D_K}$. Thus, releasing $\left(f_{D_1}, \ldots, f_{D_K}\right)$ also satisfies $(\alpha, \epsilon)$-RDP. We will show both conditional and parallel generation results in the following sections.

#### 4.4.2 Grayscale image sets: MNIST and Fashion MNIST

We first tried our method on MNIST and Fashion MNIST. Qualitatively, while all baselines are able to generate reasonable images under $(10, 10^{-5})$-DP guarantee (see Appendix D), our method indicates more visual improvements for smaller $\epsilon$ with more diversity and less artifacts, as shown in Figures 1 and 2. The quantitative comparison is summarized in Table 1. Although DPDM[8] is the only related work that is on a par with our method (both variants) when $\epsilon = 1$, DP-kernel significantly outperforms other baselines when $\epsilon = 0.2$. Specifically, our method improves the SoTA FID from $61.9$ to $26.5$ on MNIST, and from $78.4$ to $46.2$ on FMNIST, while SoTA classification accuracy is generally improved for more than $20\%$ on MNIST and more than $10\%$ on FMNIST.

The $FD_{DINOv2}$ of our conditional variant on MNIST and FMNIST are $454.8$ and $608.9$ when $\epsilon = 1$, and they are $410.3$ and $589.2$ when $\epsilon = 0.2$.

---

[8]DPDM requires 8 GPUs to train, whereas our method only requires 1 GPU.

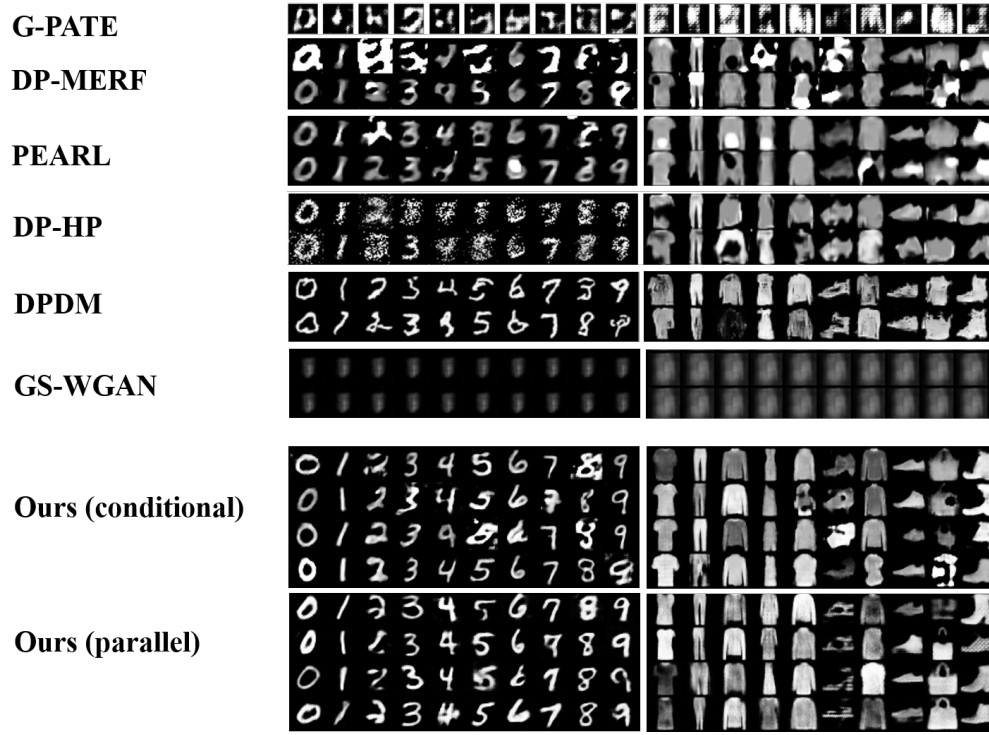

Figure 1: Qualitative comparison under $(1, 10^{-5})$-DP on MNIST and Fashion MNIST

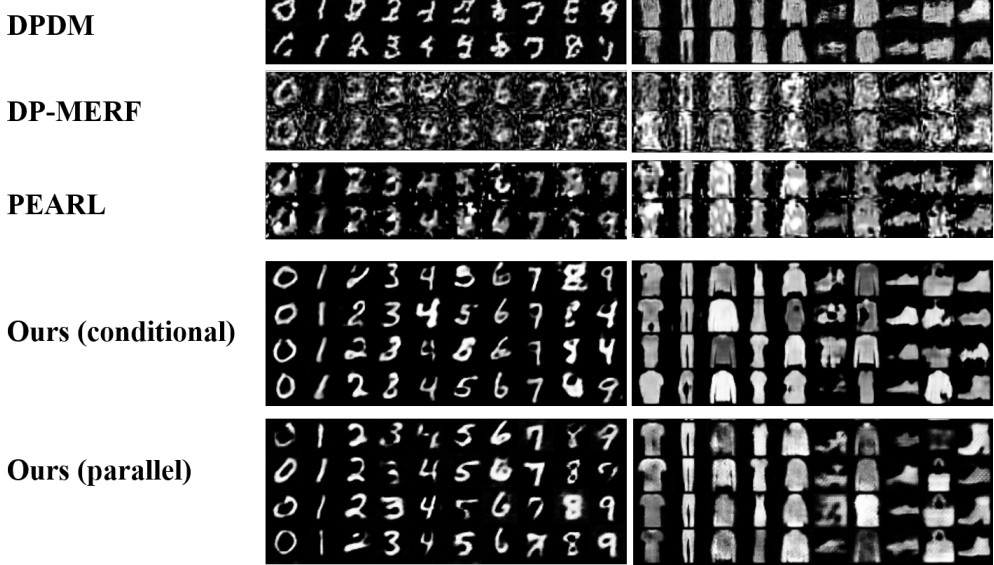

Figure 2: Qualitative comparison under $(0.2, 10^{-5})$-DP on MNIST and Fashion MNIST

### 4.4.3   RGB image sets: CelebA

To test the versatility and scalability of our method, we also evaluate our method on a more complex colorful image dataset, i.e. CelebA. Figure 3 shows that our method generates more diverse face images with identifiable gender attributes compared to baselines, which is quantitatively verified by significant improvement in both FID and Acc in Table 1. Note that DPDM is unconditional on CelebA. Notably, for $\epsilon = 0.2$, our method improves the SoTA FID from 264.8 to 85.9, and improves SoTA Acc from 52.7 to 86.5.

The FD$_{\text{DINOv2}}$ of our conditional variant on CelebA are 1028.2 when $\epsilon = 1$ and 998.9 when $\epsilon = 0.2$.

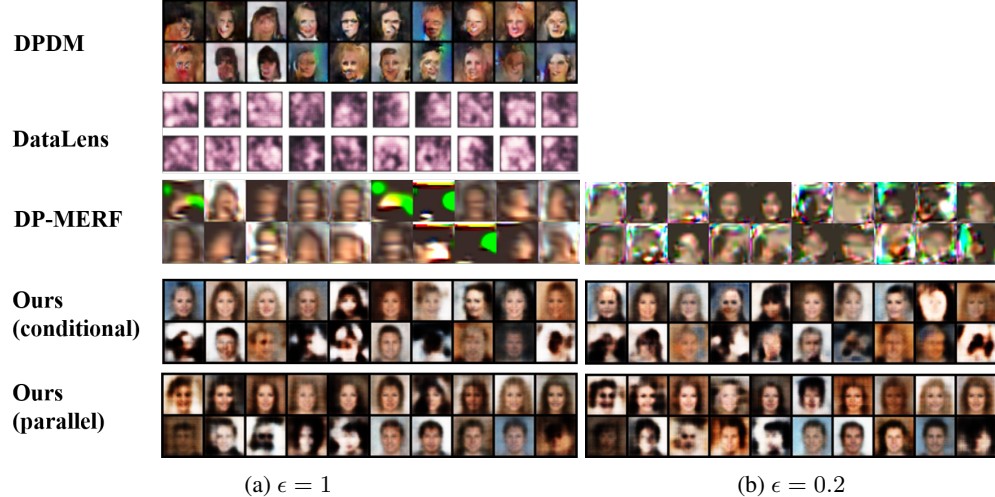

|  | (a) $\epsilon = 1$ | (b) $\epsilon = 0.2$ |

Figure 3: The qualitative comparison on CelebA. $\delta = 10^{-5}$. DPDM is unconditional.

Table 1: Quantitative comparison on MNIST, Fashion MNIST (FMNIST) and CelebA. Acc denotes classification accuracy, which is shown in %. ↑ and ↓ refer to higher is better or lower is better, respectively. We use boldface for the best performance. Results of DP-CGAN, GS-WGAN, DP-Sinkhorn are cited from Cao et al. [4] and Long et al. [21]. Results of PEARL and DPDM are cited from Dockhorn et al. [6]. Results of G-PATE and DataLens are cited from their papers, respectively. (*): DPDM did unconditional generation on CelebA.

| Method | $\epsilon$ | MNIST | | FMNIST | | CelebA | |
|---|---|---|---|---|---|---|---|
| | | FID ↓ | Acc ↑ | FID ↓ | Acc ↑ | FID ↓ | Acc ↑ |
| DP-MERF | 1 | 118.3 | 80.5 | 104.2 | 73.1 | 219.4 | 57.6 |
| GS-WGAN | 1 | 489.8 | 14.3 | 587.3 | 16.6 | 437.3 | 62.9 |
| DP-HP | 1 | - | 74.0 | - | 67.0 | - | - |
| PEARL | 1 | 121.0 | 78.2 | 109.0 | 68.3 | - | - |
| G-PATE | 1 | 153.4 | 58.8 | 214.8 | 58.1 | 293.2 | 70.2 |
| DataLens | 1 | 186.1 | 71.2 | 195.0 | 64.8 | 297.7 | 70.6 |
| DPDM | 1 | 23.4 | 93.4 | **37.8** | 73.6 | *__71.0__ | - |
| **Ours (conditional)** | 1 | 29.5 | 93.4±0.5 | 49.5 | 78.8±0.4 | **81.8** | 86.2±0.9 |
| **Ours (parallel)** | 1 | **21.8** | **95.5**±0.6 | 48.4 | **80.0**±0.5 | 83.8 | **87.0**±0.8 |
| DP-MERF | 0.2 | 119.3 | 75.2 | 151.3 | 67.4 | 264.8 | 52.7 |
| PEARL | 0.2 | 133.0 | 77.6 | 160.0 | 68.0 | - | - |
| G-PATE | 0.2 | - | 22.0 | - | 18.0 | - | - |
| DataLens | 0.2 | - | 23.4 | - | 22.3 | - | - |
| DPDM | 0.2 | 61.9 | 71.9 | 78.4 | 57.0 | - | - |
| **Ours (conditional)** | 0.2 | **26.5** | 91.3±0.8 | **46.2** | 78.4±0.7 | **85.9** | 84.3±1.1 |
| **Ours (parallel)** | 0.2 | 37.3 | **93.7**±0.7 | 65.4 | **79.2**±0.5 | 95.8 | **86.5**±1.2 |

## 5  Conclusion

We generalize RDP for vectors to functional mechanisms and develop all building blocks, e.g. (sub-sampled) Gaussian mechanisms, composition theorems, and post-processing theorems, to facilitate its adoption in deep learning frameworks. We show that those main properties or results of v-RDP also hold for f-RDP. Equipped with f-RDP, we propose a novel approach for training a DPGM, by making the loss function in the RKHS private without truncating the RKHS feature map. Experimental results across different datasets and privacy costs indicate that our method (equipped with f-RDP and retaining the full discriminative capability of the kernel) consistently outperforms other kernel-based methods (with v-RDP) as well as non-kernel-based methods by a large margin. We expect our work to bridge the gap between RDP and functional mechanisms and enrich the family of DPGM.

## Acknowledgments and Disclosure of Funding

We thank the reviewers and the area chair for thoughtful comments that have improved our final draft. YY gratefully acknowledges NSERC and CIFAR for funding support. Resources used in preparing this research were provided, in part, by the Province of Ontario, the Government of Canada through CIFAR, and companies sponsoring the Vector Institute.

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

# A   Proofs

**Theorem 2** (Parallel composition of v-RDP). *If mechanism $\mathcal{M}_i$ satisfies $(\alpha, \epsilon_i)$-RDP for $i = 1, 2, \ldots, m$, and let $D_1, D_2, \ldots, D_m$ be the disjoint partitions by executing a deterministic partitioning function $P$ on $D$. Releasing $\mathcal{M}_1(D_1), \ldots, \mathcal{M}_m(D_m)$ satisfies $(\alpha, \max_{i \in \{1,2,\ldots,m\}} \epsilon_i)$-RDP.*

*Proof.* Without loss of generality, given two neighboring datasets $D$ and $D'$, assume that $D$ contains one more element than $D'$. Executing $P$ on $D$ and $D'$, we have partitions $D_1, D_2, \ldots, D_m$ and $D'_1, D'_2, \ldots, D'_m$, respectively. There exists $j$ such that (1) $D_j$ contains one more element than $D'_j$, and (2) $D_s = D'_s$ for $s = 1, 2, \ldots, m$ and $s \neq j$. Denote $\mathcal{M}_1(D_1), \ldots, \mathcal{M}_m(D_m)$ by $\mathcal{M}(D)$. Using additivity of Rényi divergence in [30] (Thm 28):

$$
\begin{aligned}
\mathbb{D}_\alpha(\mathcal{M}(D)||\mathcal{M}(D')) &= \sum_{i=1}^{m} \mathbb{D}_\alpha(\mathcal{M}_i(D_i)||\mathcal{M}_i(D'_i)) \\
&= \sum_{i=1,\ldots,m, i \neq j} \mathbb{D}_\alpha(\mathcal{M}_i(D_i)||\mathcal{M}_i(D'_i)) + \mathbb{D}_\alpha(\mathcal{M}_j(D_j)||\mathcal{M}_j(D'_j)) \\
&\leq \epsilon_j \leq \max_{i=1,2,\ldots,m} \epsilon_i.
\end{aligned}
$$

The proof is complete. $\qquad\square$

**Theorem 3** (Post-processing theorem of f-RDP). *If a function $f_D$ is $(\alpha, \epsilon)$-RDP, so is $g \circ f_D$, where $g$ is a post-processing mechanism that only depends on a finite number of outputs of $f_D$.*

*Proof.* Given any finite subsets $S$, we reach the reduction of proof of the post-processing theorem in [23]:

$$
\mathbb{D}_\alpha(f_D(S)||f_{D'}(S)) \geq \mathbb{D}(g(f_D(S))||g(f_{D'}(S))).
$$

$\qquad\square$

**Lemma 1.** *Definition 4 implies:*

$$
\mathbb{P}(\widetilde{f_D} \in A) \leq \left( \exp(\epsilon)\mathbb{P}(\widetilde{f_{D'}} \in A) \right)^{\frac{\alpha-1}{\alpha}}, \quad \forall A \in \mathcal{F}_0, \tag{9}
$$

*where we reuse all the notations in Definition 3.*

*Proof.* Proof sketch: we will show that Eq. (9) holds whenever Eq. (4) holds.

By taking logarithm and rearranging from Eq. (9), we have

$$
\frac{1}{\alpha-1} \log \left( \left( \frac{\mathbb{P}(\widetilde{f_D} \in A)}{\mathbb{P}(\widetilde{f_{D'}} \in A)} \right)^\alpha \mathbb{P}(\widetilde{f_{D'}} \in A) \right) \leq \epsilon. \tag{10}
$$

By the definition of cylinder sets, $\widetilde{f_D} \in A$ implies that $\widetilde{f_D}(S)$ is in some sets $B$ for any finite subsets $S = (\mathbf{x}_1, \ldots, \mathbf{x}_n)$ of $T$. Thus,

$$
\mathbb{P}(\widetilde{f_D} \in A) = \mathbb{P}(\widetilde{f_D}(S) \in B) = \int_S p(x)\, \mathrm{d}\mu(x).
$$

where $p$ is the density of $\widetilde{f_D}(S)$. Now we can translate Eq. (10) into

$$
\frac{1}{\alpha-1} \log \left( \left( \frac{\int_S p(x)\, \mathrm{d}\mu(x)}{\int_S q(x)\, \mathrm{d}\mu(x)} \right)^\alpha \int_S q(x)\, \mathrm{d}\mu(x) \right) \leq \epsilon. \tag{11}
$$

Recall that Eq. (4) in Definition 4 can be translated to:

$$
\frac{1}{\alpha-1} \log \left( \int (\frac{p(x)}{q(x)})^\alpha q(x)\, \mathrm{d}\mu(x) \right) \leq \epsilon.
$$

Note that $p, q$ are non-negative, so $\int (\frac{p(x)}{q(x)})^\alpha q(x)\,\mathrm{d}\mu(x) \geq \int_S (\frac{p(x)}{q(x)})^\alpha q(x)\,\mathrm{d}\mu(x)$. Compared to Eq. (11), it now suffices to show

$$\int_S (\frac{p(x)}{q(x)})^\alpha q(x)\,\mathrm{d}\mu(x) \geq \Big(\frac{\int_S p(x)\,\mathrm{d}\mu(x)}{\int_S q(x)\,\mathrm{d}\mu(x)}\Big)^\alpha \int_S q(x)\,\mathrm{d}\mu(x).$$

Define $p^*(x) = \frac{p(x)\mathbb{1}(x\in S)}{\int_S p(x)\,\mathrm{d}\mu(x)}$, and $q^*$ is similarly defined. All we want to show reduces to

$$\int (\frac{p^*(x)}{q^*(x)})^\alpha q^*(x)\,\mathrm{d}\mu(x) \geq 1 \iff \mathbb{E}_{x\sim q^*}[(\frac{p^*}{q^*})^\alpha] \geq \big(\mathbb{E}_{x\sim q^*}[\frac{p^*}{q^*}]\big)^\alpha = 1,$$

where the final step follows from Jensen's inequality. $\qquad\square$

**Proposition 3** (f-RDP conversion to f-DP). *A function $\widetilde{f_D}$ that is $(\alpha, \epsilon)$-RDP is $(\epsilon + \frac{\log 1/\delta}{\alpha-1}, \delta)$-DP.*

*Proof.* We reuse the notations in Definition 3 in this proof. To show that an $(\alpha, \epsilon)$-RDP function satisfies $(\epsilon', \delta)$-DP for functions, where $\epsilon' = \epsilon + \frac{\log 1/\delta}{\alpha-1}$, the objective becomes to show $\mathbb{P}[\widetilde{f_D} \in A] \leq \exp(\epsilon') \times \mathbb{P}[\widetilde{f_{D'}} \in A] + \delta$. With the help of Lemma 1, denote $\mathbb{P}[\widetilde{f_{D'}} \in A]$ by $Q$ and we perform a case analysis:

- If $\exp(\epsilon)Q \leq \delta^{\alpha/(\alpha-1)}$, then

$$\mathbb{P}(\widetilde{f_D} \in A) \leq \big(\exp(\epsilon)Q\big)^{\frac{\alpha-1}{\alpha}} \leq \delta \leq \exp(\epsilon')Q + \delta.$$

- If $\exp(\epsilon)Q > \delta^{\alpha/(\alpha-1)}$, then

$$\begin{aligned}
\mathbb{P}(\widetilde{f_D} \in A) &\leq \big(\exp(\epsilon)Q\big)^{\frac{\alpha-1}{\alpha}} \\
&= \exp(\epsilon)Q\big(\exp(\epsilon)Q\big)^{-\frac{1}{\alpha}} \\
&\leq \exp(\epsilon)Q \cdot \delta^{-\frac{1}{\alpha-1}} \\
&= \exp(\epsilon + \frac{\log 1/\delta}{\alpha-1})Q \\
&\leq \exp(\epsilon')Q + \delta.
\end{aligned}$$

Combining the two cases completes the proof. $\qquad\square$

**Theorem 4** (Parallel composition of f-RDP). *Given a deterministic partitioning function $P$, let $D_1, D_2, \ldots, D_m$ be the disjoint partitions by executing $P$ on $D$. If function $f_{D_i}$ satisfies $(\alpha, \epsilon_i)$-RDP for $i = 1, 2, \ldots, m$, releasing $(f_{D_1}, \ldots, f_{D_m}) := f_D$ satisfies $(\alpha, \max_{i\in\{1,2,\ldots,m\}} \epsilon_i)$-RDP.*

*Proof.* Without loss of generality, given two neighboring datasets $D$ and $D'$, assume that $D$ contains one more element than $D'$. Executing $P$ on $D$ and $D'$, we have partitions $D_1, D_2, \ldots, D_m$ and $D'_1, D'_2, \ldots, D'_m$, respectively. There exists $j$ such that (1) $D_j$ contains one more element than $D'_j$, and (2) $D_s = D'_s$ for $s = 1, 2, \ldots, m$ and $s \neq j$. Given any finite subsets $S = (\mathbf{x}_1, \ldots, \mathbf{x}_n) \subset T$, Using additivity of Rényi divergence in [30] (Thm 28), we have:

$$\begin{aligned}
\mathbb{D}_\alpha(f_D(S)\|f_{D'}(S)) &= \sum_{i=1}^m \mathbb{D}_\alpha(f_{D_i}(S)\|f_{D'_i}(S)) \\
&= \sum_{i=1,\ldots,m,i\neq j} \mathbb{D}_\alpha(f_{D_i}(S)\|f_{D'_i}(S)) + \mathbb{D}_\alpha(f_{D_j}(S)\|f_{D'_j}(S)) \\
&\leq \epsilon_j \leq \max_{i\in\{1,2,\ldots,m\}} \epsilon_i.
\end{aligned}$$

$\qquad\square$

**Theorem 5** (Sequential composition of f-RDP). *Let $\{f_D : D \in \mathcal{D}\}$ and $\{g_D : D \in \mathcal{D}\}$ be two families of functions indexed by dataset $D$, where $f_D \in \mathcal{R}_1^T$ is $(\alpha, \epsilon_1)$-RDP and $g_D : \mathcal{R}_1^T \to \mathcal{R}_2^S$ is $(\alpha, \epsilon_2)$-RDP. Releasing the sequentially composed functional mechanism $h_D = (f_D, g_D \circ f_D) \in \mathcal{R}_1^T \times \mathcal{R}_2^S = (\mathcal{R}_1 \times \mathcal{R}_2)^{T \times S}$ satisfies $(\alpha, \epsilon_1 + \epsilon_2)$-RDP.*

*Proof.* According to Definition 4, our objective is to show for any finite subsets $X = (\mathbf{x}_1, \ldots, \mathbf{x}_n)$ of $T$ and $Y = (\mathbf{y}_1, \ldots, \mathbf{y}_m)$ of $S$,

$$\mathbb{D}_\alpha(h_D(X,Y) \| h_{D'}(X,Y)) \leq \epsilon_1 + \epsilon_2$$

$$\iff \mathbb{D}_\alpha\Big(\big((f_D(X), g_D(f_D, Y))\big) \| \big(f_{D'}(X), g_{D'}(f_{D'}, Y)\big)\Big) \leq \epsilon_1 + \epsilon_2.$$

Here we adapt the proof of Proposition 1 in [23]. Let $F$ be the distribution of $f_D(X)$, $G$ be the distribution of $g_D(f_D, Y)$, $H = (F, G)$, and $F', G', H'$ are similarly defined on adjacent dataset $D'$.

$$
\begin{aligned}
\exp[(\alpha - 1)\mathbb{D}_\alpha(h_D(X,Y) \| h_{D'}(X,Y))] &= \int_{\mathcal{R}_1 \times \mathcal{R}_2} H(x,y)^\alpha H'(x,y)^{1-\alpha}\, \mathrm{d}x\, \mathrm{d}y \\
&= \int_{\mathcal{R}_1} \int_{\mathcal{R}_2} [F(x)G(x,y)]^\alpha [F'(x)G'(x,y)]^{1-\alpha}\, \mathrm{d}x\, \mathrm{d}y \\
&= \int_{\mathcal{R}_1} F(x)^\alpha F'(x)^{1-\alpha} \Big[ \int_{\mathcal{R}_2} G(x,y)^\alpha G'(x,y)^{1-\alpha}\, \mathrm{d}y \Big]\, \mathrm{d}x \\
&\leq \int_{\mathcal{R}_1} F(x)^\alpha F'(x)^{1-\alpha} \cdot \exp[(\alpha-1)\epsilon_2] \\
&\leq \exp[(\alpha-1)(\epsilon_1 + \epsilon_2)].
\end{aligned}
$$

$\square$

**Proposition 4** (Gaussian mechanism for f-RDP). *Let $G$ be a sample path of a Gaussian process having mean zero and covariance function $k$. Let $M$ denote the Gram matrix (as defined in Eq. (2)). Let $\{f_D : D \in \mathcal{D}\}$ be a family of functions indexed by database $D$. Releasing $\widetilde{f_D} = f_D + \sigma \Delta \cdot G$ satisfies $(\alpha, \frac{\alpha}{2\sigma^2})$-RDP whenever Eq. (3) holds.*

*Proof.* Consider any finite set $(\mathbf{x}_1, \ldots, \mathbf{x}_n) \in T^n$, the vector $(G(\mathbf{x}_1), \ldots, G(\mathbf{x}_n))$ follows a multivariate Gaussian with mean zero and covariance given by Eq. (2). Thus, evaluating $\widetilde{f_D}$ at any finite sets would form a vector that satisfies Eq. (5) in Definition 5, which completes the proof. $\square$

# B    Datasets

**MNIST [15] & Fashion MNIST [34]:**   MNIST contains hand-written digits images, whereas Fashion MNIST contains cloth and shoe images. Images in both datasets are single-channel, in the size of $1 \times 28 \times 28$, which are resized to $1 \times 32 \times 32$ and normalized to have $0.5$ mean and $0.5$ standard deviation. Both datasets have 10 classes. We adopt the official training and test split. MNIST and Fashion MNIST are made available under Creative Commons Attribution-Share Alike 3.0 license and MIT License, respectively.

**CelebA [20]:**   CelebA is a dataset including face images of celebrities. Each image is in the size of $3 \times 178 \times 218$ and has 40 binary attributes. All images are center-cropped to $3 \times 178 \times 178$, then resized to $3 \times 32 \times 32$, and normalized to have $0.5$ mean and $0.5$ standard deviation. We also adopt the official training, validation and test split, but randomly select 60k images from the training split as our training set. The CelebA dataset is available for non-commercial research purposes only, as described on their website.

# C    Implementation

**Generative network:**   Our unconditional generative network is based on the official *code* of MMD-GAN [16]. For the NN architecture, we use the same Conv2dTranpose layer with similar depth as prior related works. Besides, we use the same architecture for both gray-scale and RGB image sets to

Table 2: Training parameters for retaining DP on MNIST and Fashion MNIST. Both variants on CelebA use parameters in the row of Conditional. $q$ is the subsampling rate, and $\sigma$ is the noise multiplier.

|  | target $\epsilon$ | $q$ | $\sigma$ | epochs (iterations) |
|---|---|---|---|---|
| Conditional | 10.0 | 0.001 | 0.60 | 200 (200k) |
|  | 1.0 | 0.001 | 1.95 | 200 (200k) |
|  | 0.2 | 0.001 | 8.00 | 200 (200k) |
| Parallel | 10.0 | 0.010 | 1.00 | 200 (20k) |
|  | 1.0 | 0.010 | 5.75 | 200 (20k) |
|  | 0.2 | 0.010 | 25.00 | 200 (20k) |

demonstrate its versatility (of course with different NN parameters to account for the difference in input channels). Encoding labels to the latent code leads to the conditional variant. All networks are optimized by RMSprop with a learning rate $5 \times 10^{-5}$.

**CNN classifier:**    We follow Cao et al. [4] for the classifier implementation.

The CNN consists of following layers: Conv2d($input\_channels$, 32, kernel_size=3, stride = 2, padding=1) $\rightarrow$ Dropout(p=0.5) $\rightarrow$ ReLU $\rightarrow$ Conv2d(32, 64, kernel_size=3, stride = 2, padding=1) $\rightarrow$ Dropout(p=0.5) $\rightarrow$ ReLU $\rightarrow$ flatten $\rightarrow$ linear($flatten\_dim$, $output\_dim$) $\rightarrow$ Softmax.

The CNN classifier is optimized by Adam with default parameters. All classifiers are trained on synthetic data, and we report test accuracy on real test data as the evaluation metric.

**Privacy:**    We use *Tensorflow privacy* for computing the total privacy cost, which only requires inputting a few important parameters, e.g. subsampling rate (or batch size), noise multiplier, training epochs (iterations), target $\delta$ ($\delta = 10^{-5}$ in all our experiments). We summarize the parameters in Table 2.

**Baselines:**    We use the official code of DP-MERF to replicate their results under different $\epsilon$ and use the same evaluation metrics to add quantitative comparison. The numerical evaluations of rest baselines are cited from related works, as specified in the caption of the tables.

**Computation resources:**    All computation is conducted by one NVIDIA T4 GPU. It costs 1.5 hours to train a conditional generator on MNIST (and Fashion MNIST) and around 5 hours on CelebA. Each sub-model in the parallel variant takes only 10 minutes to train on MNIST (and Fashion MNIST) and 30 minutes on CelebA. A better GPU should compute faster. As a reference, DP-Sinkhorn and GS-WGAN can take up to 24 hours to train, and GS-WGAN even requires at least 2 GPUs (as they have 1000 discriminators).

# D    Additional results

For completeness, we include the qualitative comparison between our method and related works under $(10, 10^{-5})$-DP on all three datasets in Figures 4 and 5. Numerical comparison is in Table 3. We note that $\epsilon = 10$ is usually considered a weak privacy regime. Generally, all baselines can generate decent images when $\epsilon = 10$, whereas our method still generates more diverse and more informative images, especially on CelebA.

# E    Broader impact and limitations

**Broader impact:**    Some prior works reveal that DP training could introduce additional biases to the model, leading to fairness concerns [2, 10, 35]. We did not explore this point in the current experiments as there are no sensitive attributes involved, but we can imagine the potential negative effects on fairness issues if our methods are not used properly.

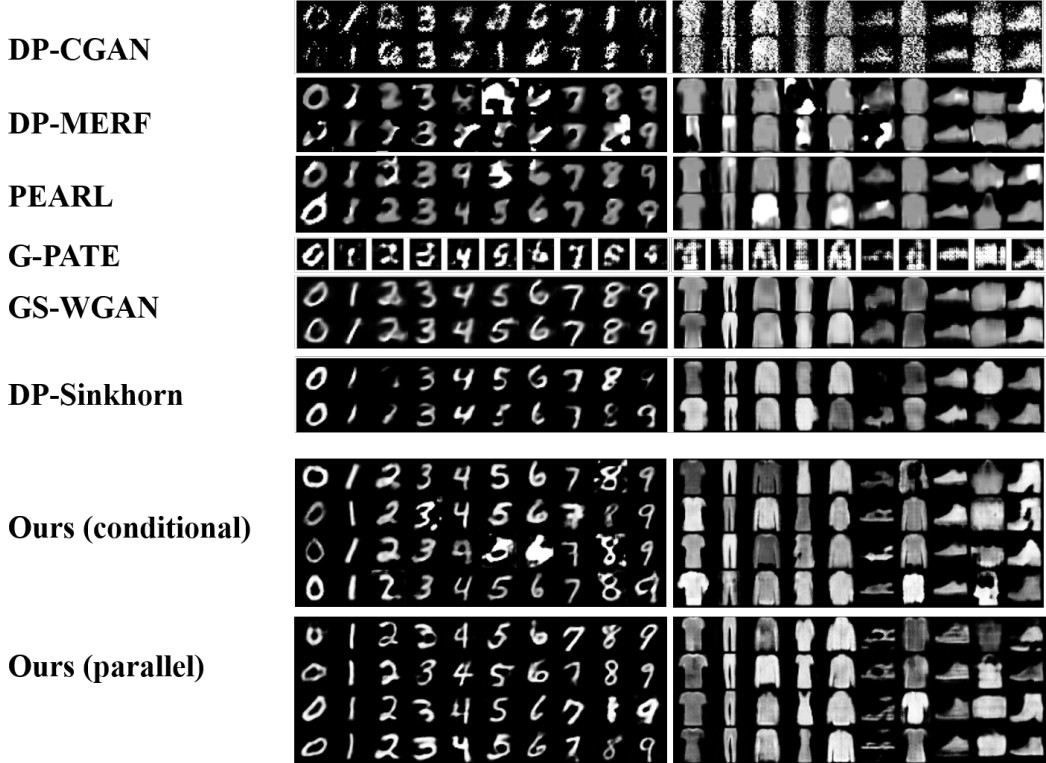

Figure 4: Qualitative comparison under $(10, 10^{-5})$-DP on MNIST and Fashion MNIST

Table 3: Quantitative comparison under $(10, 10^{-5})$. Acc denotes classification accuracy, which is shown in %. ↑ and ↓ refer to higher is better or lower is better, respectively. Results of DP-CGAN, GS-WGAN, DP-Sinkhorn are cited from Cao et al. [4] and Long et al. [21]. Results of PEARL and DPDM are cited from Dockhorn et al. [6]. Results of G-PATE and DataLens are cited from their papers, respectively. (*): DPDM did unconditional generation on CelebA.

| Method | $\epsilon$ | MNIST | | FMNIST | | CelebA | |
|---|---|---|---|---|---|---|---|
| | | FID ↓ | Acc ↑ | FID ↓ | Acc ↑ | FID ↓ | Acc ↑ |
| DPDM | 10 | 5.0 | 97.3 | 18.6 | 84.9 | *21.1 | - |
| DP-CGAN | 10 | 179.2 | 63.0 | 243.8 | 46.0 | - | - |
| DP-MERF | 10 | 121.4 | 82.0 | 110.4 | 73.2 | 211.1 | 64.0 |
| PEARL | 10 | 116.0 | 78.8 | 102.0 | 64.9 | - | - |
| G-PATE | 10 | 150.6 | 80.9 | 171.9 | 69.3 | 305.9 | 70.7 |
| DataLens | 10 | 173.5 | 80.7 | 167.7 | 70.6 | 320.8 | 72.9 |
| GS-WGAN | 10 | 61.3 | 80.0 | 131.3 | 65.0 | 432.5 | 63.3 |
| DP-Sinkhorn | 10 | 55.6 | 79.1 | 129.4 | 68.9 | - | - |
| **Ours (conditional)** | 10 | 29.9 | 93.1 | 56.3 | 79.2 | 77.2 | 85.8 |
| **Ours (parallel)** | 10 | 17.7 | 96.4 | 38.1 | 82.0 | 78.5 | 87.4 |

Besides, we do not expect this work to trigger any differential privacy concern by its design, and we do not expect it to be used for Deepfakes (as the non-private counterparts are better in utility), but there might be some unintended uses that we are not aware of for now.

**Limitations:** The current conditional approach assumes a uniform distribution over different labels, which happens to hold for all three datasets. A more tailored algorithm is needed if the label distribution is imbalanced. The current parallel approach is not subject to a specific label distribution, but it may hinder its practical deployment if the number of classes is too large.

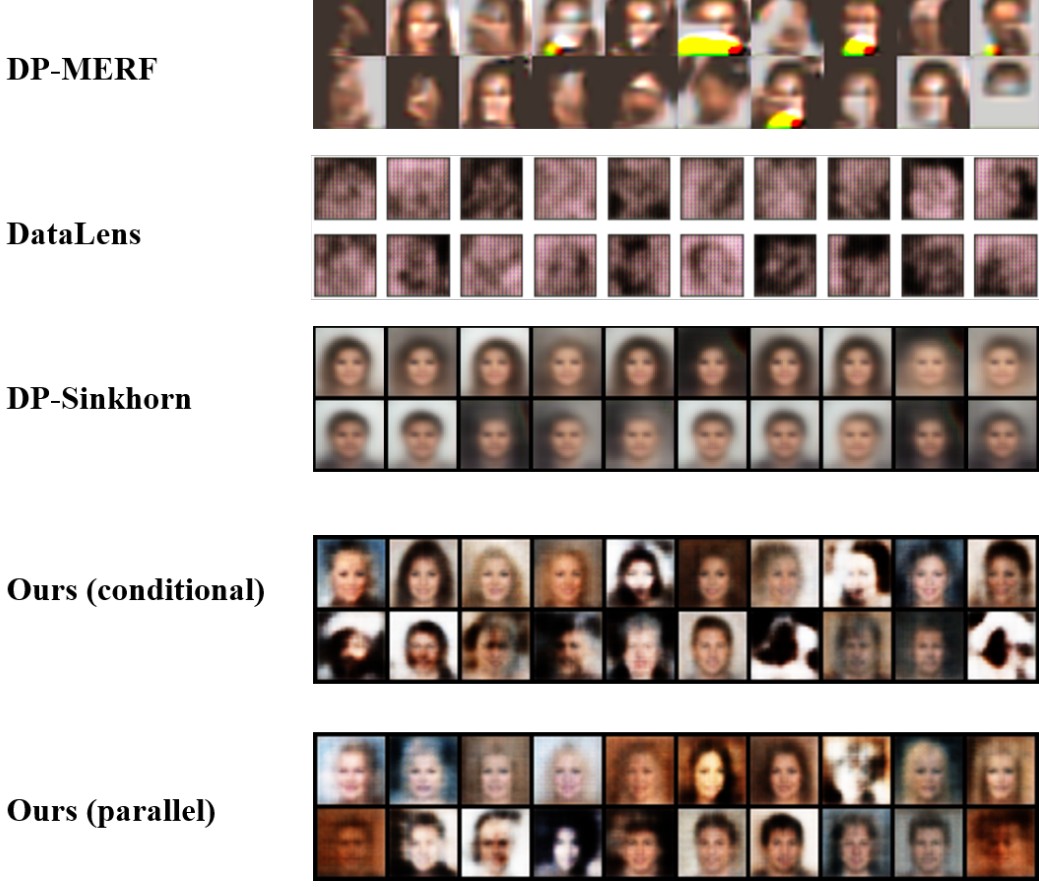

**DP-MERF**

**DataLens**

**DP-Sinkhorn**

**Ours (conditional)**

**Ours (parallel)**

Figure 5: Qualitative comparison under $(10, 10^{-5})$-DP on CelebA

## F A simplified view of RDP for vectors and functions

In this section, we propose to define Rényi differential privacy (RDP) for vectors and functions in a unified way.

Let $\mathcal{D}$ be a collection of datasets and $f : \mathcal{D} \to \mathcal{R}$ be a *randomized* mechanism. We consider $\mathcal{R} = \mathbb{R}^T$ so we may also identify $f$ as a bivariate (random) function $f : T \times \mathcal{D} \to \mathbb{R}$. In the following we call two datasets $D, D' \in \mathcal{D}$ adjacent if they differ by one data point. We recall the Rényi $\alpha$-divergence [26] between two random variables $\mathsf{X}_1$ and $\mathsf{X}_2$ with density $p$ and $q$, respectively:

$$\mathbb{D}_\alpha(\mathsf{X}_1 \| \mathsf{X}_2) := \mathbb{D}_\alpha\big(p(\mathbf{x}) \| q(\mathbf{x})\big) := \tfrac{1}{\alpha-1} \log \mathbb{E}_{\mathsf{X} \sim q} \left[ \tfrac{p(\mathsf{X})}{q(\mathsf{X})} \right]^\alpha \geq 0,$$

where $\alpha > 1$ is a hyperparameter and the last inequality follows from Jensen's inequality. Since we can rewrite

$$\mathbb{D}_\alpha(\mathsf{X}_1 \| \mathsf{X}_2) = \tfrac{1}{\alpha-1} \log \mathbb{E}_{\mathsf{X} \sim p} \left[ \tfrac{p(\mathsf{X})}{q(\mathsf{X})} \right]^{\alpha-1},$$

it is clear Rényi's $\alpha$-divergence $\mathbb{D}_\alpha$ is increasing w.r.t. $\alpha$, with the following limits:

$$\mathsf{KL}(\mathsf{X}_1 \| \mathsf{X}_2) = \lim_{\alpha \to 1} \mathbb{D}_\alpha(\mathsf{X}_1 \| \mathsf{X}_2) \leq \lim_{\alpha \to \infty} \mathbb{D}_\alpha(\mathsf{X}_1 \| \mathsf{X}_2) = \operatorname*{ess\,sup}_{\mathbf{x}} \log \tfrac{p(\mathbf{x})}{q(\mathbf{x})}.$$

For joint distributions, we can easily verify the following decomposition rule:

$$\begin{aligned}
\mathbb{D}_\alpha(p(\mathbf{x}, \mathbf{z}) \| q(\mathbf{x}, \mathbf{z})) &= \tfrac{1}{\alpha-1} \log \mathbb{E}_{\mathsf{X} \sim q} \left[ \left[ \tfrac{p(\mathsf{X})}{q(\mathsf{X})} \right]^\alpha \exp\left( (\alpha-1) \mathbb{D}_\alpha\left[ p(\mathbf{z}|\mathsf{X}) \| q(\mathbf{z}|\mathsf{X}) \right] \right) \right] \\
&\geq \mathbb{D}_\alpha\big(p(\mathbf{x}) \| q(\mathbf{x})\big),
\end{aligned} \tag{12}$$

where the inequality follows from the nonnegativity of the conditional Rényi divergence inside the exponential. In other words, the $\alpha$-divergence between joint distributions is always larger than that between marginal distributions.

We are now ready to define Rényi differential privacy [23]:

**Definition 6** (($\alpha, \epsilon$)-RDP). *A randomized mechanism $f : T \times \mathcal{D} \to \mathbb{R}$ is ($\alpha, \epsilon$)-RDP if for all adjacent datasets $D, D' \in \mathcal{D}$,*

$$\sup_{n \in \mathbb{N}} \sup_{\mathbf{t} := [t_1, \ldots, t_n] \subseteq T} \mathbb{D}_\alpha\big(f(\mathbf{t}; D) \| f(\mathbf{t}; D')\big) \leq \epsilon, \tag{13}$$

*where $f(\mathbf{t}; D) := [f(t_1; D), \ldots, f(t_n; D)]$ is the concatenation of $n$ random variables.*

The acute reader will notice some immediate difference between our definition and the one in the literature, e.g., Mironov [23, Definition 4]. Before explaining our motivation, let us first notice that when $T = [d] := \{1, \ldots, d\}$, i.e., the range of our mechanism $f$ is $\mathcal{R} = \mathbb{R}^d$ (which is the setting of [23] and most existing work), the supremum in (13) is attained when $n = d$ and $\mathbf{t} = [1, \ldots, d]$, thanks to inequality (12). In other words, the supremum in (13) is superficial when $T$ is a finite set. The important point is that we can now extend RDP effortlessly to any index set $T$, including when $T$ is even uncountable. For instance, in our later application to generative modeling (Section 4), we will use $T = \mathbb{R}^m$. On the other hand, one might be tempted to lump all random variables together into $f(T; D) = \{f(t; D) : t \in T\}$, hoping to obviate the supremum in (13). However, this approach will run into two difficulties: Firstly, the rigorous minds will start to worry about measurability issues around $f(T; D)$ (as a stochastic process) and involve tedious measure-theoretic jargon (such as the cylinder field in [12]). Secondly, we no longer possess a dominating measure (e.g., Lebesgue) to define an easily computable density of the process $f(T; D)$. In contrast, our reduction to the finite case through taking the supremum does not suffer from either issue, and, as we will see, renders extensions of existing properties of RDP (to an arbitrary index set $T$) rather natural or even trivial.

The following result was proved by Mironov [23, Proposition 3]:

**Theorem 6.** *If a mechanism $f$ is ($\alpha, \epsilon$)-RDP, then for any (measurable) set $A$,*

$$\Pr[f(D) \in A] \leq \{\exp(\epsilon) \Pr[f(D') \in A]\}^{1 - \frac{1}{\alpha}} \leq (1 - \tfrac{1}{\alpha}) \exp(\epsilon + \tfrac{1}{\alpha - 1} \log \tfrac{1}{\delta}) \Pr[f(D') \in A] + \tfrac{\delta}{\alpha}.$$

*In particular, $f$ is $(\epsilon + \tfrac{1}{\alpha - 1} \log \tfrac{1}{\delta}, \tfrac{\delta}{\alpha})$-DP.*

*Proof.* For $\alpha \geq 1$, consider the (jointly) convex function (as the perspective of $p \mapsto p^\alpha$):

$$g(p, q) := q \left(\tfrac{p}{q}\right)^\alpha,$$

as well as the probability measure $\mathbf{1}_{\mathbf{x} \in A} \cdot \mu(\mathrm{d}\mathbf{x}) / \mu(A)$ (recall that $\mu(\mathrm{d}\mathbf{x})$ is the underlying dominating measure that the densities $p$ and $q$ are defined w.r.t.). Applying Jensen's inequality we obtain

$$\tfrac{1}{\mu(A)} \cdot \int_A q(\mathbf{x}) \left(\tfrac{p(\mathbf{x})}{q(\mathbf{x})}\right)^\alpha \mu(\mathrm{d}\mathbf{x}) \geq g\left(\tfrac{1}{\mu(A)} \int_A p(\mathbf{x}) \mu(\mathrm{d}\mathbf{x}), \tfrac{1}{\mu(A)} \int_A q(\mathbf{x}) \mu(\mathrm{d}\mathbf{x})\right)$$

$$= \tfrac{1}{\mu(A)} Q(A) \left(\tfrac{P(A)}{Q(A)}\right)^\alpha,$$

where $P(A) := \int_A p(\mathbf{x}) \mu(\mathrm{d}\mathbf{x}) = \Pr[f(D) \in A]$ and similarly for $Q(A)$. Thus,

$$\epsilon \geq \mathbb{D}_\alpha(p \| q) \geq \tfrac{1}{\alpha - 1} \log \int_A q(\mathbf{x}) \left(\tfrac{p(\mathbf{x})}{q(\mathbf{x})}\right)^\alpha \mu(\mathrm{d}\mathbf{x}) \geq \tfrac{1}{\alpha - 1} \log \left[Q(A) \left(\tfrac{P(A)}{Q(A)}\right)^\alpha\right].$$

Rearranging we obtain the first inequality. The second inequality is simply the arithmetic-geometric mean inequality, after noting that

$$[\exp(\epsilon) Q(A)]^{1 - \frac{1}{\alpha}} = \left[\exp(\epsilon + \tfrac{1}{\alpha - 1} \log \tfrac{1}{\delta}) Q(A)\right]^{1 - \frac{1}{\alpha}} \cdot \delta^{\frac{1}{\alpha}}.$$

Lastly, we relax the factor $(1 - \tfrac{1}{\alpha}) \leq 1$ to obtain the claimed DP guarantee. $\qquad\square$

For two Gaussian distributions $\mathcal{N}(\mathbf{u}, \Sigma)$ and $\mathcal{N}(\mathbf{v}, \Sigma)$, their Rényi $\alpha$-divergence can be readily computed from Liese and Vajda [18, Proposition 2.22]:

$$\mathbb{D}_\alpha \left( \mathcal{N}(\mathbf{u}, \Sigma) \| \mathcal{N}(\mathbf{v}, \Sigma) \right) = \tfrac{\alpha}{2} \left\langle \mathbf{u} - \mathbf{v}, \Sigma^{-1}(\mathbf{u} - \mathbf{v}) \right\rangle.$$

Therefore, the Gaussian mechanism

$$f(D) := F(D) + \sigma \cdot \mathcal{N}(\mathbf{0}, \Sigma)$$

is $(\alpha, \epsilon)$-RDP, where

$$\epsilon = \tfrac{\alpha}{2\sigma^2} \cdot \sup_{D \sim D'} \left\langle F(D) - F(D'), \Sigma^{-1}(F(D) - F(D')) \right\rangle.$$

