# OpenReview forum: "Functional Renyi Differential Privacy for Generative Modeling"
_NeurIPS.cc/2023/Conference — NeurIPS 2023 poster_

### Official Review · Reviewer_1DXa · 2023-07-04

**Soundness:** 4 excellent
**Presentation:** 3 good
**Contribution:** 3 good
**Rating:** 7
**Confidence:** 3

**Summary:**

Standard differential privacy (v-DP) provides mathematical guarantees on the closeness of distributions of real-valued, randomized functions when they act on adjacent inputs. v-RDP is a variant of the standard DP definition which can be connected back to v-DP guarantees, and can provide tighter v-DP bounds through its analysis. Meanwhile, functional DP (f-DP) extends v-DP to the case where randomized functionals act on adjacent input sets of functions. The current paper considers an RDP variant of f-DP (f-RDP). From the proposed definition of f-RDP, the authors present and prove a number of theorems that mirror the standard toolkit used by practitioners for v-DP and v-RDP, namely (serial and parallel) composition, post-processing, the subsampled Gaussian mechanism, and conversion to f-DP. f-RDP is used to analyze the training of private generative models in several experiments.

**Strengths:**

Thank you to the authors for sharing their interesting research! This work contains both theoretical results, proving several useful theorems following from the definition of f-RDP, and experimental work training generative models with privacy guarantees. The theoretical work appears to be correct, and I have read over (but not verified in a detailed manner) the proofs given in the appendix. The experimental results show improvement in image quality at strong levels of privacy ($\epsilon=0.2$) on three common datasets.

Originality: The application of functional DP to generative modelling is relevant and timely.

Quality: The analysis appears to be correct, the usual toolkit of theorems used by DP practitioners is filled out, and at least some of the results are non-trivial to obtain. Experiments and code are provided.

Clarity: The format and flow of the paper are generally good. Proofs are reasonably detailed.

Significance: The possibility of useful generative models trained at privacy levels as strong as $\epsilon=0.2$ is very promising.

**Weaknesses:**

Originality: The paper combines two known concepts - RDP, and f-DP. The main theoretical results play out exactly as one would expect - the authors do not go beyond the standard toolkit to explore any new properties of their definition.

Quality: Quantitative results are given as single numbers in Table 1 and 3 - there is no quantification of statistical uncertainty. Please see questions below for more suggestions to improve quality.

Clarity: The paper assumes a high level of background mathematical knowledge which may not be accessible to the ML community, and does not include pedagogical expanations. Example topics that are not explained in detail, or for which intuition is not provided: functionals, Borel sets from measure theory, cylinder sets, reproducing kernel Hilbert spaces, Gaussian processes. The lack of explanation and intuition will limit the readership of this work.

Significance: Only a few examples of the usefulness of differentially private functionals are given (L29-31). The applications of f-RDP seem limited in comparison to v-RDP. Only relatively small and easy datasets are used in the experiments.

**Questions:**

1. Can the authors quantify the improvement in $\epsilon$ when using the f-RDP analysis as compared to the preexisting f-DP analysis? For comparison, the motivation to use v-RDP is that it produces tighter guarantees v-DP than have been obtained with v-DP directly.

2. Beyond providing a mathematical guarantee, have the authors considered what characteristics DPGMs will have in comparison to non-private GMs? Do they behave differently in terms of training or sample generation?

3. Related to the previous question, can the authors verify (beyond the mathematical guarantees) that their DPGMs preserve privacy of the training data? For example, around the Figure 1,2 experiments, have the authors considered checking nearest neighbours between the private training data, and the generated samples? The same could be done for generated samples from non-private models. I would expect that models trained with DP guarantees should not memorize and reproduce training set data. Such verification would give more intuitive meaning to the levels of privacy considered in the work ($\epsilon=0.2, 1.0, 10.0$). An alternate test would be to perform membership inference attacks.

4. L286 and L292 mention standard deviations of results, but the method of obtaining these statistics is not mentioned. Can the authors include explanations of where the statistics come from? E.g. are models retrained from scratch with different randomness? How many times? Can error bars be included for the main results tables (Table 1 and Table 3 in the appendix)? Specifically for the Accuracy measure where 5 classifiers were trained, why are error bars withheld?

The chosen evaluation metrics have significant flaws and limitations. I encourage the authors to put more thought into the metrics they choose:

5. FID - Concerns around FID have been widely raised in the DGM literature, but the authors make no mention of this. The authors state FID is a measure of generation fidelity which is only partly true, as FID also measures diversity and these two factors cannot be disentangled. Can the authors comment on why they believe FID is a sufficient measure, or provide alternative evaluations? Note that using a measure "because it is the standard" may not be a good reason if that measure has significant flaws.

6. Accuracy of a classifier trained on generated samples and evaluated on real samples - This is not a typical measure, as it would only seem to measure if generation has picked up on class-conditional information. It seems to have been used in [3], but this is not mentioned in the main text (only the appendix). Additionally, having to train a model (actually five!) on a generated dataset just to get an evaluation seems like an expensive procedure, and one that introduces additional randomness (on top of the randomness in generating a dataset for evaluation). Imagine trying to use this measure for validation during training! At the very least, the authors must explain what this metric is measuring and why they believe this to be a useful measure for their task.

Minor:

L2 and L21 refer to RDP as a "recent" development. While recency is always relative, work from 2017 does not seem very recent with the fast pace of development in ML.

The proof of Theorem 4 follows the proof of Theorem 2. However, Theorem 4's proof lacks the same detail as Theorem 2's. Can the authors either provide additional steps for Theorem 4, or make a reference back to Theorem 2? The reader who only looks at Theorem 4 may feel there is insufficient detail.

A bit more exposition on the experimental details in the main text would be good to contextualize the results, for instance what kind of NN architectures are used, and are these similar across training methods for fair comparisons?

There are some grammatical errors that do not detract from the clarity of the work (non-exhaustive list):

L2 becomes -> has become

L18 becomes -> has become

L226 pluggin -> plugging

L230 Wlog - this abbreviation is somewhat unprofessional.

**Limitations:**

The consideration of broader impacts in Appendix E seems like an afterthought. I encourage the authors to put more thought into this section. The paper focuses on the theoretical, and utility aspects of DP, but does not mention that private training can have negative effects other than reducing utility. For instance, it is well-known that DP-SGD usually exacerbates the biases in models making them more unfair [A][B][C]. How does training with the loss functions in Section 4 compared to DP-SGD in terms of fairness? This is an important "broader impact" of research into private model training.

[A] Bagdasaryan et al. 'Differential privacy has disparate impact on model accuracy' NeurIPS 2019

[B] Xu et al. 'Removing disparate impact on model accuracy in differentially private stochastic gradient descent' ACM SIGKDD 2021

[C] Esipova et al. 'Disparate impact in differential privacy from gradient misalignment' ICLR 2023

----
Summary of Discussion: I have read all reviews and rebuttals for this submission.

The authors have responded to each of my concerns. The only outstanding piece was around the evaluation metrics chosen by the authors, which they admitted had flaws. Due to the format of NeurIPS reviews (no updated manuscript) I can't fully review the changes they intend to make.

---

> ### Author Rebuttal · Authors · 2023-08-09
>
> Due to the length limit, we will simplify the reviewer's question when we quote.
> > Q1. Originality: The paper combines RDP and f-DP, and does not explore any new properties of proposed f-RDP.
>
> **Response:** We believe that we have developed the necessary tools for the practical use of f-RDP, and we demonstrate its value through a GM example (where the current tools are sufficient). We hope that the f-RDP toolkit in this work will encourage and facilitate more practitioners to exercise functional f-RDP in real applications, and we certainly look forward to exploring more properties in future work.
>
> > Q2. Quality: There is no error bar in Table 1 and 3; L286 and L292 mention std of results, can the authors include explanations of the statistics? E.g. are models retrained from scratch? Can error bars be included in the tables ?
>
> **Response:** The model is not repetitively trained. Once the model is trained, we randomly generate samples 5 times and run the classification accordingly, then we can compute the mean and std of Acc. Note that we do not know the comparison to baselines, as they generally do not report the error bars in their tables. Nevertheless, we will include our std in our tables.
>
> We also tried retraining the model from scratch on MNIST, and run the same classification tasks 5 times, where both mean and variance are very similar to what we reported in Table 1. For example, when $\epsilon=1,0.2$, our Acc are $93.2\pm0.6$ and $91.8\pm0.5$, respectively.
>
> > Q3. Clarity: The paper assumes a high level of background knowledge.
>
> **Response:** We will add more explanations and pointers in our revision to make it more reader-friendly.
>
> > Q4. Significance: Only a few examples of functional DP are given (L29-31). The applications of f-RDP seem limited in comparison to v-RDP. Only relatively small and easy datasets are used in the experiments.
>
> **Response:** There were a few more examples of f-DP, but comparatively, yes, there are more v-DP papers than f-DP, partly because v-RDP/v-DP are sufficient for many purposes and partly because of the technical requirements and the lack of functional DP tools. We believe the f-RDP mechanism provides more flexibility (see our joint response to CQ1) and we hope that our work would help other practitioners to adopt f-RDP in more applications.
>
> As for the datasets, both MNISTs and CelebA are widely compared image benchmarks in related works, so we feel it is more fair to compare the results using the same datasets. We agree that bigger and more complex datasets are interesting and useful, and we currently leave it for future work.
>
> > Q5. Can the authors quantify the improvement in $\epsilon$ when using the f-RDP as compared to the preexisting f-DP? For comparison, v-RDP produces tighter v-DP guarantees than obtained with v-DP directly.
>
> **Response:** We have shown that the conversion from f-RDP to f-DP is the same as from v-RDP to v-DP (Proposition 3). Therefore, the improvement in $\epsilon$ of f-RDP over f-DP is the same as that of v-RDP over v-DP.
>
> > Q6. What characteristics DPGMs will have in comparison to non-private GMs? Do they behave differently?
>
> **Response:** The DPGM is expected to be not too different by changing one point in the training set, so that the individual privacy is preserved. Intuitively, because of the noise perturbation during training, we expect the generation from DPGMs will be less similar to the training set compared to non-private GMs (they are expected to reproduce/memorize fewer training samples than non-private GMs).
>
> > Q7. CQ2.
>
> **Response:** Please see our joint response to CQ2.
>
> > Q8. Concerns around FID have been widely raised in the DGM literature. The authors' statement on FID is only partly true.
>
> **Response:** We do note that FID is a combined measure for fidelity and diversity. We give a concise description due to page limit. We will add more notes on FID in our revision.
>
> In spite of its flaw, FID is still one of the most widely used evaluation metrics in image generation tasks, which you can find in both recent non-private and private GM papers. We understand that FID is flawed, but we fear that it will raise more questions if we fail to include FID in our comparison.
>
> > Q10. The authors must explain what Acc is measuring and why they believe this is a useful measure for their task.
>
> **Response:** Classification accuracy is actually used by all DPGM works, as a quantification of model utility. It is measured by training a classifier on 60k generated samples, then we test the classifier on the real test set. It simulates the scenario of how a generative model can be used: generating synthetic samples for downstream ML tasks, which is one of the main motivations for developing DPGMs.
>
> > Q11. Minor comments:
> > + Q11.1. RDP is not 'recent'.
>
> **Response:** We will modify our term.
>
> > + Q11.2. Theorem 4's proof lacks details.
>
> **Response:** We will include more details in it.
>
> > + Q11.3. Need more details on the experiments, e.g., NN architecture.
>
> **Response:** For the NN architecture, we use the same Conv2dTranpose layer with similar depth as prior related works. Besides, we use the same architecture for both gray-scale and RGB image sets to demonstrate its versatility (of course with different NN parameters to account for the difference in input channels). We will add more details about our experiments in our revision.
>
> > + Q11.4. There are some grammatical errors.
>
> **Response:** We will proofread it again carefully.
>
> > Q12. The paper does not mention that private training can have fairness issues [A][B][C]. This is an important "broader impact".
>
> **Response:** Thanks for your suggestion. We agree this is a very important and interesting direction. We did not explore this point in the current experiments as there are no sensitive attributes involved, but we can imagine the potential negative effects if our methods are not used properly. We will include this discussion in our broader impact.

---

> > ### Comment · Reviewer_1DXa · 2023-08-13
> > **Response to Rebuttal**
> >
> > Thank you to the authors for providing responses to my questions and concerns. I have read all reviews and rebuttals.
> >
> > > Q8: In spite of its flaw, FID is still one of the most widely used evaluation metrics in image generation tasks, which you can find in both recent non-private and private GM papers. We understand that FID is flawed, but we fear that it will raise more questions if we fail to include FID in our comparison.
> >
> > I understand that FID is probably the most widely used measure of performance for generative models, but that does not imply that it is a good measure, or sufficient for understanding the quality of generated samples in this work. I agree with the authors that not including FID could raise questions, and for instance prevent easy comparison to related work. However, including FID does not prevent the authors from also including better evaluation metrics. This could help the community break out of a cycle of using flawed metrics, simply because they are "widely used".
> >
> > I do not want to dictate to the authors exactly which metrics they should use, but I encourage them to put more thought into the evaluation of their models, and explore the wide literature on this topic. As a starting point I could recommend a recent paper that reviews and compares many such metrics [D]. (I note that [D] first appeared after the NeurIPS submission deadline, but many of the references therein have been available for years.
> >
> > [D] Stein et al. "Exposing flaws of generative model evaluation metrics and their unfair treatment of diffusion models"

---

> > > ### Author Response · Authors · 2023-08-16
> > >
> > > Thanks for your constructive suggestion! We agree that FID is flawed and read through the reference the reviewer mentioned.
> > >
> > > As pointed out by [D], FID is found to be inconsistent with human evaluation, partly because the Inception-V3 network is not good enough. The authors found that using the same Frechet Distance (FD) but with a more advanced feature extractor, e.g. DINOv2 ViT-L/14, will work better (more consistent with human judgment). So we took some time to compute a new FD_DINOv2 for our method using their code base, and we report our result in the following table. The metric is computed from real test images and 10k generated images.
> > >
> > >
> > >
> > > |                | MNIST | FMNIST | CelebA |
> > > |----------------|-------|--------|--------|
> > > | $\epsilon=1$   | 454.8 | 608.9  | 1028.2 |
> > > | $\epsilon=0.2$ | 410.3 | 589.2  | 998.9  |
> > > |                |       |        |        |

---

### Official Review · Reviewer_i7xA · 2023-07-05

**Soundness:** 2 fair
**Presentation:** 2 fair
**Contribution:** 2 fair
**Rating:** 5
**Confidence:** 4

**Summary:**

The work generalizes the DP to infinite dimensions. Based on the modeling of Hall et al., the authors propose some characteristics of functional DP aligning with the traditional DP. The most interesting part is adopting their proposed framework to RHKS for an infinite functional space. Some numerical studies is done for verification.

**Strengths:**

Pros:
1. the application in RKHS is interesting and promising for further application to reduce the dimension and formulate some functional structural data input
2. some proposed tools like composition and subsampling can be adopted for further studies

**Weaknesses:**

Cons:
1. The technical contribution should be clearly stated. Some remarks about the different characteristics of the functional DP and traditional Dp in Subsample and others should be stated. We do know whether we directly imply the results for the composition theorem with Hall's and the traditional DP's proof.
2. Some more error bounds for RKHS need to be considered. It might influence the privacy guarantee.
3. The work does not clearly show the complexity result to construct such a DP measure. We can do it with kernel trick or reduced paramters, but some guarantee should be provided.

**Questions:**

1. How can we compute the functional DP given function $f$, if the functional is high dimension, will we suffer from dimension curse
2. what is the relationship between the proposed DP and the traditional DP. This problem comes from numerical studies. I am not sure whether it is fair to use the same $epsilon$ for different methods with different privacy metrics. Maybe they protect different things.
3. can functional form be generalized to privacy loss random variable accounting


**Limitations:**

Yes.

---

> ### Author Rebuttal · Authors · 2023-08-09
>
> >  Q1. The technical contribution should be clearly stated. Some remarks about the different characteristics of the functional DP and traditional Dp in Subsample and others should be stated.
>
> **Response:** Thanks for your suggestion. Our primary technical contribution is the functional RDP with its toolkit, e.g., subsampling, composition theorems, and Gaussian mechanisms. Note that there are no subsampling and composition theorems in Hall et al.'s work.
>
> The proposed functional RDP extends traditional RDP from vectors to functions, where the output space can be infinite-dimensional. We have indicated that the functional RDP shares similar results with traditional RDP in subsampling, composition, conversion to $(\epsilon,\delta)$-DP, Gaussian mechanisms.
>
> > Q2. Some more error bounds for RKHS need to be considered. It might influence the privacy guarantee.
>
> **Response:**
> Thanks for your suggestion. This seems to be a delicate issue that is worth significant future study. We will give a preliminary result below (akin to Proposition 3.1 of DP-MERF), based on a finite dimensional argument. Assuming the generated samples $w$ are fixed, so that we may restrict the kernel to a finite domain $X = \\{x_1, \ldots, x_N, w_1, \ldots, w_M\\}$. Then, let
>
> $$ F = \Big\lVert \frac{1}{N} \sum_{i=1}^N k(\cdot, x_i) - \frac{1}{M} \sum_{j=1}^M k(\cdot, w_j) \Big\rVert_{X}^2 = \lVert\mu_P(\cdot)-\mu_Q(\cdot) \rVert_X^2 $$
>
> $$\tilde F = \Big\lVert \mathbf{g} + \tfrac{1}{N} \sum_{i=1}^N k(\cdot, x_i) - \tfrac{1}{M} \sum_{j=1}^M k(\cdot, w_j) \Big\rVert_X^2 = \lVert\mathbf{g} + \mu_P(\cdot) -\mu_Q(\cdot) \rVert_X^2,$$
>
> where $\mu_P(\cdot),\mu_Q(\cdot)$ are embeddings of real distribution $P$ and generative distribution $Q$, $\lVert\cdot\rVert_X$ denotes the RKHS norm restricted to the domain $X \times X$, and $\mathbf{g}$ is a Gaussian sample path with zero mean and covariance $\Sigma$. Note that $F$ is the standard empirical objective of maximum mean discrepancy (MMD) while $\tilde F$ is the privatized version of $F$. We can bound the difference between $F$ and $\tilde F$:
>
> $$ \mathbb{E} _\mathbf{g}|\tilde F- F| \leq  \mathbb{E} _\mathbf{g}[\lVert\mathbf{g}\rVert_X^2 + 2\lVert\mathbf{g}\rVert_X \cdot \lVert\mu_P(\cdot)-\mu_Q(\cdot)\rVert_X] \leq \mathbb{E} _\mathbf{g}[\lVert\mathbf{g}\rVert_X^2]  + 2 \sqrt{\mathbb{E} _\mathbf{g} \lVert\mathbf{g}\rVert_X^2} \cdot \lVert\mu_P(\cdot)-\mu_Q(\cdot)\rVert_X. $$
>
> Using the RKHS norm equality over a finite domain (see, e.g., Saitoh and Sawano 2016, p. 130), we have
>
> $$\mathbb{E} _\mathbf{g}\big[\lVert\mathbf{g}\rVert_X^2\big] = \mathbb{E} _\mathbf{g}\big[\mathbf{g}^\top K^{-1} \mathbf{g}\big]  = trace(\Sigma K^{-1}) = \sigma^2 \Delta^2 (N+M),$$
>
> where $K = k(X, X)$ is the kernel matrix and $\Sigma = \sigma^2 \Delta^2 K$. Noting that the kernel $k$ is bounded by (say) 1, we further have
>
> $$ \Delta^2 \leq \frac{2}{N^2}, ~ \mbox{ and } \lVert\mu_P(\cdot)-\mu_Q(\cdot)\rVert_X \leq 2. $$
>
> Setting $M=N$ (for simplicity) and combining all of the above, we obtain
>
> $$\mathbb{E}_{\mathbf{g}} | \tilde F - F | \leq \frac{4\sigma^2}{N} + \frac{8\sigma}{\sqrt{N}}.$$
>
> Thus, as $N=M$ increases, the difference between the privatized and non-privatized objectives decreases at the usual rate of $O(1/\sqrt{N})$. We note that this result is similar to Lemma B.3 of DP-MERF (effectively equating their dimension $D$ with sample size $m$). However, since we do not truncate the feature map of the kernel, we do not suffer the additional truncation error in their Lemma B.2.
>
> > Q3. The work does not clearly show the complexity result to construct such a DP measure.
>
> **Response:** The time complexity for one iteration is $O\big((M+N)^2\big)$ (where $N$ is the size of private training data and $M$ is the size of generated samples).
>
> > Q4. How can we compute the functional DP given function $f$, if the functional is high dimension, will we suffer from dimension curse.
>
> **Response:** We only considered real-valued functions in this work. Our Proposition 4 shows how we can make a general function $f$ DP with Gaussian process, and Corollary 2 is a special case where the function is in an RKHS. Since the parameters $\sigma$ and $\Delta$ in Corollary 2 do not depend on the input dimension (explicitly), we will not suffer from the dimension curse. We believe this is inherited from the nice properties of kernel mean embedding.
>
> > Q5. what is the relationship between the proposed DP and the traditional DP. This problem comes from numerical studies. I am not sure whether it is fair to use the same $\epsilon$ for different methods with different privacy metrics. Maybe they protect different things.
>
> **Response:** The proposed f-RDP generalizes traditional v-RDP from finite-dimensional vectors to infinite-dimensional functions. Our Section 3 shows that the main properties of f-RDP are the same as v-RDP, and we already know how to convert v-RDP to v-DP. Therefore, the numerical procedure for computing the $\epsilon$ for f-RDP is the same as traditional DP. In all of our experiments, we  compute and convert the DP guarantee to the same $(\epsilon,\delta)$-DP. Thus, we believe the comparison is wrt the same privacy metric, which is fair. Besides, both f-RDP and traditional DP protect the same individual privacy, i.e. the functions or vectors will not change much by changing one point in the dataset.
>
> > Q6. can functional form be generalized to privacy loss random variable accounting
>
> **Response:** If we understand correctly, the privacy loss random variable referred by the reviewer is from Abadi et al.'s work (DP-SGD) and is associated with the Moments Accountant (for traditional v-DP) in the same paper.
>
> Yes. We have already shown that the f-RDP accounting is the same as v-RDP, which can be easily converted to v-DP. Thus, we believe the functional form can be generalized to other DP notions / accountants, including the privacy loss random variable accounting (Moments Accountant).

---

> > ### Comment · Reviewer_i7xA · 2023-08-19
> >
> > The authors have addressed some of my concerns. I decide to keep my score.

---

### Official Review · Reviewer_iWpL · 2023-07-05

**Soundness:** 2 fair
**Presentation:** 2 fair
**Contribution:** 2 fair
**Rating:** 5
**Confidence:** 4

**Summary:**

The authors  extend  the concept of RDP from vectors  to functions. They introduce the notion of functional RDP and present a few properties of the functional RDP under composition / amplification by sampling. Finally, they demonstrate the functional RDP using the application in RKHS.

**Strengths:**

1. They generalize the definition of RDP to the setting when we have an infinite output space.
2. The application to RKHS is nice and the experiment section is also complete.

**Weaknesses:**

My main concern is the novelty of the paper: The applications on DP generative models and the proposed functional new RDP definition appear to publish either the fixed finite-dimensional vector output or the infinite output with some strong constraint on its boundness.  These aspects have already been covered in the existing RDP literature.

Though the authors claim the goal of their work is to publish a function, their results essentially involve releasing the output of functions  applied on a private dataset. This algins closely with  prior classic DP work.

**Questions:**

Here I list my concerns on the novelty:


1. Their results on RDP for functions (Definition 5)  and subsampled RDP for functions directly generalize from the existing  RDP literature, except that the output space might now have an infinite-dimension. However,  assuming a bounded RDP divergence of the unbounded output space (in this work) essentially renders it equivalent to  classic RDP.

2. The authors provide empirical demonstrations of the applications of functional RDP by applying it to privatize the Maximum Mean Discrepancy (MMD), a crucial component in DP generative model training. In their approach, they choose to privatize the kernel function $k(\cdot, \cdot)$ applied to the private dataset, resulting in a finite (and even one-dimensional) output space for the function.
Given that the output space is finite, the authors could use the classic Gaussian mechanism directly to publish the loss function. In this context, the connection with functional RDP is not apparent.

Some minor comments: The organization of the paper is not clear. The authors do not clearly state the motivation/applications of the functional  RDP  before proving their main results.

**Limitations:**

Not applicable.

---

> ### Author Rebuttal · Authors · 2023-08-09
>
> > Q1. My main concern is the novelty of the paper: Their results on RDP for functions (Definition 5) and subsampled RDP for functions directly generalize from the existing RDP literature, except that the output space might now have an infinite-dimension. However, assuming a bounded RDP divergence of the unbounded output space (in this work) essentially renders it equivalent to classic RDP.
>
> **Response:** Thanks for your comment. Our novelty lies in the following two aspects: (1) Theoretically, **we are the first to develop functional RDP**, and we also develop many useful tools to facilitate its practical use; (2) Empirically, we propose a novel DPGM training algorithm with functional RDP, which allows us to perturb the training objective without truncating the RKHS, in contrast to previous works that truncate RKHS to a finite-dimensional space and add standard Gaussian noise (but lose injectivity of the kernel mean embedding). Our experimental results on widely compared image benchmarks consistently outperform previous related works, especially in strict privacy regimes (e.g. $\epsilon=1,0.2$).
>
> > Q2. The authors provide empirical demonstrations of the applications of functional RDP by applying it to privatize the Maximum Mean Discrepancy (MMD), a crucial component in DP generative model training. In their approach, they choose to privatize the kernel function $k(\cdot,\cdot)$ applied to the private dataset, resulting in a finite (and even one-dimensional) output space for the function. Given that the output space is finite, the authors could use the classic Gaussian mechanism directly to publish the loss function. In this context, the connection with functional RDP is not apparent.
>
> **Response:** If we understand correctly, the reviewer is saying that our $\widetilde{f_D}$ is evaluated at finitely many points (including both real and generated data), then recalling equation (8) in our paper, we are effectively releasing a finite vector $[\widetilde{f_D}(x_1), \ldots, \widetilde{f_D}(x_N), \widetilde{f_D}(w_1(\theta)), \ldots, \widetilde{f_D}(w_M(\theta))]$. Therefore, we can apply the classic Gaussian mechanism (we suppose the reviewer refers to the method in DP-MERF) to this vector. However, while the (private) training data is fixed (if we treat $N$ as its size), the generated data $w(\theta)$ will increase with training iterations (and is not determined _a priori_; it changes with the model parameter $\theta$), i.e., its size is $tM$ where $t$ denotes the number of iterations and $M$ is the number of generated samples in each iteration, which could effectively lead to a large sensitivity due to a large $t$ and is likely to degrade the model utility. Certainly, we can pre-set an upper bound for $tM$ (the total number of generated samples), but it may restrict the generalizability of the model  and we still would not know the values of generated data beforehand (and hence would have to guard against the worst-case in calculating the sensitivity).
>
> Alternatively, we can consider a more flexible approach via releasing the function $\widetilde{f_D}$ with DP guarantees since a function can be evaluated at arbitrarily many points. To do so, we need a functional DP mechanism, which is what we propose in this paper, i.e., we apply functional RDP to the function $\widetilde{f_D}$ by adding a sample path of a Gaussian process, which does not depend on the number of generated samples anymore.
>
> > Q3. Some minor comments: The organization of the paper is not clear. The authors do not clearly state the motivation/applications of the functional RDP before proving their main results.
>
> **Response:** Thanks for your suggestion. We mentioned the motivation/applications at the beginning of Introduction (Line 24-31), and we will follow your suggestion to expand it and make it more clear in our revision.

---

> > ### Comment · Reviewer_iWpL · 2023-08-20
> >
> > Thank you for the clarification. My concern on Q2 is addressed and I have raised my score.
> > I have another following question: DP-MERF needs a composition rule to analyze the privacy loss over train iterations. Will the privacy loss of your approach increase when we increase training iterations?

---

> > > ### Author Response · Authors · 2023-08-20
> > >
> > > Thanks for your question. When DP-MERF releases the privatized KME of a subsampled batch in each training iteration, their privacy loss will increase with training iterations and they need composition. In this case, our method works in a similar manner: subsampling a batch to index the function, then applying functional DP mechanism to make the function DP, so yes, the privacy cost of our approach will increase with training iterations, and that's why we need subsampled Gaussian mechanism (Sec 3.6) and sequential composition (Sec 3.4).

---

> > > > ### Comment · Reviewer_iWpL · 2023-08-21
> > > >
> > > > Thank you for the clarification.

---

### Official Review · Reviewer_JRvJ · 2023-07-09

**Soundness:** 3 good
**Presentation:** 3 good
**Contribution:** 4 excellent
**Rating:** 7
**Confidence:** 3

**Summary:**

Conceptually, this paper combines the idea behind RDP and f-DP and proposes f-RDP.

Theoretically, this paper proved several important properties of f-RDP (e.g., Composition, post-processing, and the privacy level for the widely-used Gaussian mechanism).

Experimentally, this paper showed f-RDP can be applied to improve the performance of real-world applications.

**Strengths:**

1. The problem studied in this paper is important to the community. Achieving privacy without losing too much utility is always desired by researchers and mechanism designers.

2. The results of this paper look solid to me. The composition theorem for a new privacy notion usually is not an easy task.

3. The overall presentation looks good to me.

**Weaknesses:**

1. Some high-level explanations are missing and it's hard for the readers out of DP community to understand Section 2.

To both authors and chairs: I provide a relatively low confidence score because I am not an expert on generative models.

**Questions:**

1. May I know what is the high-level idea of why $f$-DP is better than DP? I guess this is because the sensitivity $\Delta$ for $f$-DP is smaller, but again, why? Can the authors provide some high-level ideas and explanations?

2. I am not an expert on generative models, but may I know why the proposed $f$-DP notions are suitable for generative models? Can it also be used to improve standard classification tasks?

3. Functional and cylinder set need to be explained before use. I don't believe the majority of the researchers in computer science (without expertise in math) can understand these.

4. I suggest the author to add a note about the $f$-DP notion in the following paper (already with 200+ citations), where $f$ plays a similar role as the Renyi divergence in RDP. I know that these two $f$-DP notions are different, but some readers may get confused because RDP is a special case of the $f$-DP in the following paper.

Dong, Jinshuo, Aaron Roth, and Weijie J. Su. "Gaussian differential privacy." arXiv preprint arXiv:1905.02383 (2019).

**Limitations:**

No limitations within my scope.

---

> ### Author Rebuttal · Authors · 2023-08-09
>
> > Q1. Some high-level explanations are missing and it's hard for the readers out of DP community to understand Section 2.
>
> **Response:** Thanks for your suggestion. We will add more explanations and summarize a paragraph in the beginning of Section 2.
>
> > Q2. May I know what is the high-level idea of why f-DP is better than DP? I guess this is because the sensitivity $\Delta$ for f-DP is smaller, but again, why? Can the authors provide some high-level ideas and explanations?
>
> **Response:** Please note that we are not claiming f-DP is always better than v-DP. They are just two DP notions for different cases (e.g., finite- or infinite-dimensional output). However, through a DPGM example, we show that privatizing the infinite-dimensional functional output in RKHS with functional RDP is theoretically more convenient and empirically better than privatizing a truncated finite-dimensional RKHS with v-DP/v-RDP.
>
> Besides, the functional DP mechanism is more flexible than traditional DP. Please see our joint response to CQ1 for details.
>
> Sensitivity is part of the reason. Since the functional output in our example is infinite-dimensional, bounding its L2 sensitivity (in the v-RDP case) will easily go to $\infty$, which will lead to infinitely large Gaussian noise and thereby is not useful. With functional RDP, we can obtain a finite sensitivity in RKHS norm. In terms of utility gain, compared to truncating the kernel, we believe that preserving the Gaussian kernel without truncation will better preserve its discriminative power. Our generation result is indeed more informative and diverse than prior related works.
>
> > Q3. I am not an expert on generative models, but may I know why the proposed f-DP notions are suitable for generative models? Can it also be used to improve standard classification tasks?
>
> **Response:** Because the MMD objective for training a generative model includes kernel functions in RKHS, we can apply f-RDP (v-RDP does not work without truncating the RKHS) therein to make the training objective differentially private. We are not aware of an immediate example in image classification, but if the training objective also includes a functional output, then we can apply f-RDP as well. It is our hope that the functional RDP toolkit detailed in this work will encourage and facilitate more practitioners to exercise functional RDP in real applications.
>
> > Q4. Functional and cylinder set need to be explained before use. I don't believe the majority of the researchers in computer science (without expertise in math) can understand these.
>
> **Response:** Thanks for your suggestion. We will add more explanations on these concepts in our revision.
>
> > Q5. I suggest the author to add a note about the $f$-DP notion in the following paper (already with 200+ citations), where $f$ plays a similar role as the Renyi divergence in RDP. I know that these two $f$-DP notions are different, but some readers may get confused because RDP is a special case of the  $f$-DP in the following paper.
>
> **Response:** Thanks for your suggestion. Indeed, the name $f$-DP may cause confusion with other notions in the field. We will follow your suggestion to change it to the more explicit functional DP.

---

> > ### Comment · Reviewer_JRvJ · 2023-08-18
> >
> > I have read the authors' response, which addressed most of my concerns. I will keep my score because the bar for "Strong Accept" is very high for NeurIPS.

---

> > > ### Author Response · Authors · 2023-08-19
> > >
> > > Thank you for your response!

---

### Author Rebuttal · Authors · 2023-08-09

We thank all reviewers for their insightful reviews. There is a doubt on the value of the proposed functional-RDP, especially in the DPGM application, e.g., why we need functional-RDP and why it is better than v-DP in this application. Besides, reviewer 1DXa raised a practical and interesting question: the different behaviors between DPGM and non-private GM in terms of generation. Therefore, we give a global response as follows for all reviewers' benefit:

> CQ1 (by reviewer iWpL). The authors provide empirical demonstrations of the applications of functional RDP by applying it to privatize the Maximum Mean Discrepancy (MMD), a crucial component in DP generative model training. In their approach, they choose to privatize the kernel function $k(\cdot,\cdot)$ applied to the private dataset, resulting in a finite (and even one-dimensional) output space for the function. Given that the output space is finite, the authors could use the classic Gaussian mechanism directly to publish the loss function. In this context, the connection with functional RDP is not apparent.

**Joint response:** If we understand correctly, the reviewer is saying that our $\widetilde{f_D}$ is evaluated at finitely many points (including both real and generated data), then recalling equation (8) in our paper, we are effectively releasing a finite vector $[\widetilde{f_D}(x_1), \ldots, \widetilde{f_D}(x_N), \widetilde{f_D}(w_1(\theta)), \ldots, \widetilde{f_D}(w_M(\theta))]$. Therefore, we can apply the classic Gaussian mechanism (we suppose the reviewer refers to the method in DP-MERF) to this vector. However, while the (private) training data is fixed (if we treat $N$ as its size), the generated data $w(\theta)$ will increase with training iterations (and is not determined _a priori_; it changes with the model parameter $\theta$), i.e., its size is $tM$ where $t$ denotes the number of iterations and $M$ is the number of generated samples in each iteration, which could effectively lead to a large sensitivity due to a large $t$ and is likely to degrade the model utility. Certainly, we can pre-set an upper bound for $tM$ (the total number of generated samples), but it may restrict the generalizability of the model  and we still would not know the values of generated data beforehand (and hence would have to guard against the worst-case in calculating the sensitivity).

Alternatively, we can consider a more flexible approach via releasing the function $\widetilde{f_D}$ with DP guarantees since a function can be evaluated at arbitrarily many points. To do so, we need a functional DP mechanism, which is what we propose in this paper, i.e., we apply functional RDP to the function $\widetilde{f_D}$ by adding a sample path of a Gaussian process, which does not depend on the number of generated samples anymore.

> CQ2 (by reviewer 1DXa). Can the authors verify (beyond the mathematical guarantees) that their DPGMs preserve privacy of the training data? For example, around the Figure 1,2 experiments, have the authors considered checking nearest neighbours between the private training data, and the generated samples? The same could be done for generated samples from non-private models. I would expect that models trained with DP guarantees should not memorize and reproduce training set data. Such verification would give more intuitive meaning to the levels of privacy considered in the work ($\epsilon=0.2,1.0,10$).

**Joint response:** Thanks for your constructive suggestion! We agree that it would be more practical and intuitive to investigate the behavior of GM with or without DP guarantees. To the best of our knowledge, no prior DPGM work has done anything similar. Nevertheless, we tried generating samples under different DP guarantees and checked their nearest neighbors in the training set (measured by L2 distance). We include the visual comparisons in the attached PDF. It is clear that the non-private GM reproduces more training samples than its private counterparts. Under weak DP guarantee ($\epsilon=10$), though the DPGM somehow still reproduces some of the training samples, we can already see more difference, e.g., in digits 3 and 4. Under the strong DP guarantee ($\epsilon=0.2$), it generates more differently compared to the non-private GM, e.g., in digits 2 to 9.

---

### Decision · Program_Chairs · 2023-09-21

**Decision:**

Accept (poster)

**Comment:**

Most reviewers praised the contribution of introducing a functional version of RDP with applications to generative modeling for being interesting and timely. They also liked the solid theory and convincing experiments. One reviewer had some concerns, some of which addressed by the author response leading that reviewer to increase his/her score. In the end, all reviewers support acceptance, and this is also my recommendation.